behaviour/ecology/genetics

sperm whale, social groups, kin relationships, DNA polymorphisms

**Author for correspondence:**
Jean-Luc Jung
e-mail: jean-luc.jung@univ-brest.fr

†These authors contributed equally to the work.

# Kin relationships in cultural species of the marine realm: case study of a matrilineal social group of sperm whales off Mauritius island, Indian Ocean

Francois Sarano[1,†], Justine Girardet[2,†],
Véronique Sarano[1], Hugues Vitry[3], Axel Preud'homme[3],
René Heuzey[4], Ana M. Garcia-Cegarra[5,6],
Bénédicte Madon[7], Fabienne Delfour[8], Hervé Glotin[9],
Olivier Adam[10,11] and Jean-Luc Jung[2]

[1]Longitude 181, Valence, France
[2]Université de Brest, Muséum National d'Histoire Naturelle, CNRS, Sorbonne Université, ISYEB, Brest, France
[3]Marine Megafauna Conservation Organisation, Mauritius
[4]Label Bleu Production, Marseille, France
[5]Centro de Investigación de Fauna Marina y Avistamiento de Cetáceos, CIFAMAC, Mejillones, Chile
[6]Departamento de Ciencias Básicas, Facultad de Ciencias, Universidad Santo Tomás, Antofagasta, Chile
[7]Université de Brest, AMURE - Aménagement des Usages des Ressources et des Espaces marins et littoraux - Centre de droit et d'économie de la mer, Plouzané, France
[8]Laboratoire d'Ethologie Expérimentale et Comparée EA 4443, Université Paris 13, Sorbonne Paris Cité, Villetaneuse, France
[9]Toulon University, Aix Marseille Université, CNRS, LIS, DYNI Team, Marseille, France
[10]Sorbonne Université, CNRS, Institut Jean Le Rond d'Alembert, UMR 7190, Paris, France
[11]Institute of Neurosciences Paris-Saclay, Bioacoustics Team, CNRS UMR 9197, Université Paris Sud, Orsay, France

J-LJ, 0000-0002-8795-8056

Understanding the organization and dynamics of social groups of marine mammals through the study of kin relationships is particularly challenging. Here, we studied a stable social group of sperm whales off Mauritius, using underwater observations, individual-specific identification, non-invasive sampling and genetic analyses based on mitochondrial

sequencing and microsatellite profiling. Twenty-four sperm whales were sampled between 2017 and 2019. All individuals except one adult female shared the same mitochondrial DNA (mtDNA) haplotype—one that is rare in the western Indian Ocean—thus confirming with near certainty the matrilineality of the group. All probable first- and second-degree kin relationships were depicted in the sperm whale social group: 13 first-degree and 27 second-degree relationships were identified. Notably, we highlight the likely case of an unrelated female having been integrated into a social unit, in that she presented a distinct mtDNA haplotype and no close relationships with any members of the group. Investigating the possible matrilineality of sperm whale cultural units (i.e. vocal clans) is the next step in our research programme to elucidate and better apprehend the complex organization of sperm whale social groups.

# 1. Introduction

Many mammals live in social groups [1]. Benefits from such group formation are diverse and include enhanced foraging efficiency, mating, knowledge sharing, taking care of juveniles and defence against predators [2–4]. Yet mammal social groups' organization, complexity and stability can vary greatly between species [5]; for example, wolves (*Canis lupus*) [6] and hamadryas baboons (*Papio hamadryas*) [7] form stable groups with a complex social structure, while European bison (*Bison bonasus*) [8] and big brown bats (*Eptesicus fuscus*) [9] can present short-term fluid associations that resemble a fission–fusion model [10].

The marine realm is no exception to this diversity [11]. Social groups presenting a remarkable stability over time and generations have been highlighted in killer whales (*Orcinus orca*) [12], sperm whale (*Physeter macrocephalus*) [13], false killer whale (*Pseudorca crassidens*) [14,15] and in long- (*Globicephala melas*) [16] and short- (*Globicephala macrorhynchus*) [17,18] finned pilot whales. These social groups are mainly composed of related individuals and often depend on kin relationships. The female role is central in group formation, which is often found to group individuals linked by more or less high levels of maternal relatedness [11]. These social groups can temporally associate with other groups, with whom they can share cultural knowledge and/or kin relationships, thus forming multilevel societies (e.g. [4,17]). In contrast with these long-term groups, other species may present more temporary social structures. Bottlenose dolphins (*Tursiops truncatus*) generally have social group dynamics that follow the fission–fusion model [19] (i.e. the composition and size of social groups vary over time via the separation and re-association of subunits), but this does not preclude extremely stable groups from forming, such as that off Sein Island in the northwest of France [20]. Some migratory species establish temporary aggregations, and these could be strongly influenced by social and kin relationships, as demonstrated for beluga whales (*Delphinapterus leucas*) [21]. In humpback whale (*Megaptera novaeangliae*), the most studied baleen whale, the mother–calf pair is the more stable social group known yet it lasts no more than a year [22]. During the first year of life, calves will follow their mothers on a migration cycle to the feeding grounds before returning to the breeding area; however, this temporary bond profoundly affects the structure of baleen whale populations. This maternally directed fidelity leads, for instance, to population structure at very small geographical scale in Russian waters [23,24]. For another baleen whale species, the Australian southern right whale (*Eubalaena australis*), microsatellites allele frequencies and kin relationships were correlated with indicators of its feeding ground locations [25].

Yet at sea, studying kinship in mammal social groups is still arduous. Boat observations allow for the photo-identification of individuals via external markings (e.g. tail fluke, dorsal fin marks, scars), although it does not provide relatedness information (except for mother–calf pairs). Therefore, genetic analyses, based on the taking of tissue samples (e.g. biopsies [26]), must be paired with photo-identification efforts to truly understand the relationship between kinship and social structure in marine mammals. Despite these challenges, there has been mounting interest, over the last few decades, in the study of cetacean social groups. Much pioneering research [27] has demonstrated that the cultural transmission of knowledge and behaviour between group members occurs in some cetacean species (e.g. [4,12,24,28–30]). Non-human social learning and cultural traditions have now been clearly recognized in several mammalian, bird, fish and insect taxa [31]. However, only in few cases has it been demonstrated that, as in humans, such cultural knowledge has shaped the evolution of a given species through gene–culture coevolution. Whitehead *et al*. [32] listed these very few cases of known gene–culture coevolution in non-human species, concerning only cetaceans and birds. Further, some cetacean species present both a matrilineal social structure and low mitochondrial genetic diversity [33]. Cultural hitchhiking—the co-transmission of knowledge and/or behaviour affecting fitness and genetic variation— could represent a

plausible explanation for this low genetic diversity [34]. Such an influence of cultural traditions on the low genetic diversity of the modern human has been proposed [35].

In this respect, the exceptional social organization and cultural traditions observed in sperm whales deserve very specific attention [4,36,37]. Female sperm whales exhibit extreme philopatry and spend their entire life with their calves in their natal social group of about 10–12 individuals, often called social units, most likely displaying a matrilineal structure [38]. Different social units can share characteristic vocal repertoires made of codas (patterns of clicks), defining vocal clans whose individuals can temporally group [39]. Male sperm whales disperse from their natal group after 6–8 years, i.e. before attaining sexual maturity [40]. Although the social units of females and juveniles are generally found in warm waters at low latitudes, the males can stay on in cold waters at high latitudes to forage and feed [41,42]. Sperm whales therefore present a highly complex and multilevel social organization, one based on matrilineal stable social groups nested within vocal clans. Such vocal clans most likely originate from the cultural transmission of vocal repertoires [4].

The extent to which this complex multilevel social organization has shaped the genetic structuring of the sperm whale species has yet to be defined. Understanding the influence of kin relationships upon the organization of its social groups is therefore of paramount interest. Few empirical studies have addressed this question, and they have provided contrasting results concerning not only the matrilineality of these social groups but also the influence of kinship on group composition [43–47]. Geographical variation probably represents the first explanation for these contradictory findings, as social group structures differ markedly between the eastern Pacific and the Atlantic [48–50]. In the Pacific, sperm whale group size could be larger and its matrilineality less strict than in the Atlantic [48]. Vocal clans are sympatric in the Pacific, but allopatric in the Atlantic [39,48], though two vocals clans have been identified in the same area in the West Indies [37].

In the Indian Ocean, very few studies on sperm whales have been undertaken. Sperm whale social groups were studied near Sri Lanka and the Seychelles [50,51]. It has been suggested that social units could be smaller in size and that they more rarely aggregate with other groups than in the Pacific [50]. Around Mauritius Island, long-term boat surveys have photo-identified around one hundred sperm whales, for which photo-identifications and codas analysis suggested the existence of several possible social units and two vocal clans in the local population [52]. A genetic approach revealed that the frequency of mitochondrial DNA (mtDNA) haplotype repartitioning in the Indian Ocean reflected a geographical (regional) structure more pronounced there than in the Pacific [53]. This genetic phenomenon could be correlated to strong female philopatry at both geographical and social levels.

The *Maubydick* project was developed to bolster our knowledge of the sperm whale at a local scale around Mauritius Island. Led by the Mauritian non-governmental organization (NGO) *Marine Megafauna Conservation Organization* since 2011, *Maubydick* aims at studying sperm whales in the waters surrounding Mauritius Island using underwater observations and professional video recordings (described in Sarano *et al.* [54]). Since 2015, a scientific study has been conducted annually by the French NGO Longitude 181, from which a catalogue of sperm whale individuals was developed, based on their morphological characteristics [54]. Thirty-eight individuals are currently identified, and the year-round presence of sperm whales near Mauritius Island is confirmed [52,54]. One social group in particular has been encountered on a regular basis at the study site since 2013, and so it appears to be resident there (figure 1). In 2019, this group was composed of 17 adult females and 11 juveniles (seven males, four females), all identified and named to facilitate their identification in the field. In the last two years, this group, named Irène's group after an adult female sperm whale of the group, seems to be splitting into two subgroups, each of stable composition.

Importantly, these repeated underwater observations and identifications have enabled the collection of sloughed skin fragments precisely matched to the individuals they came from; more than 90 of these sloughed skins have been obtained from Irène's group individuals. In this study, we analysed nuclear and mitochondrial polymorphisms in the DNA extracted from all these sloughed skin fragments, defined *genetic individuals* on the basis of shared genotypes, and then compared them to the field-identified individuals to validate the protocol of underwater sloughed skin sampling. We then analysed kin relationships between all sampled sperm whales. Our study is, to our knowledge, the first in the Indian Ocean to analyse the kin relationships, matrilineality and dynamics of a sperm whale social unit.

# 2. Material and methods

## 2.1. The *Maubydick* project

Sperm whales are common off the coast of Mauritius Island (Mascarenes Islands, Indian Ocean), where sea surface and underwater observations have been carried out since 2011, in the frame of a project called

**Figure 1.** Location of the fieldwork area, on the west coast of Mauritius Island, Indian Ocean. © 2020 Google.

*Maubydick* led by the Marine Megafauna Conservation Organization (MMCO), Trou aux Biches, Mauritius Island. The study area is located on the west coast of Mauritius Island, up to 15 km off the coast between 20.465 S, 57.334 E and 19.986 S, 57.605 E (figure 1).

Since 2015, marine fieldwork has been conducted on a regular basis, consisting of at least 160 h yr$^{-1}$ of underwater observations, mainly done between February and May. According to Mauritius rules, these observations were only carried out during the morning (from 7.30 to 12.00). All underwater observations were video recorded, either with a Sony F55 4 K, a Sony EXIR HD or a GoPro Camera.

Particular attention was paid to the individual identification of sperm whales on the basis of their external morphological characteristics, as observed underwater. For each individual, an 'identity card' was established that listed all of its principal characteristics. A name was also given to each individual in preference to an anonymous alphanumeric reference, to simplify the memorization of the individuals during their study [54].

## 2.2. Collection of individual-specific sloughed skin samples from the Mauritian sperm whales

Permission to conduct the Maubydick project, including the taking of non-invasive samples, was granted by the Mauritius Prime Minister Office on the 21 February 2017.

At first detection of a sperm whale group, the boat, a 15 m cabin cruiser designed for diving, stopped *ca* 100 m away and dropped off the observers, then it moved aside and remained *ca* 200 m away from the whales. This ensured both respect for the security rules and compliance with the charter for the responsible approach and observation of marine mammals.

Snorkelers waited for the cetaceans to swim by, to identify them with certainty based on their morphological specificities. If sperm whales did not move, snorkelers slowly and carefully swam towards them. They positioned themselves close enough to see when one or more skin fragments were released, and then manually grabbed the sloughed skin released (only one skin fragment at a time, figure 2 and electronic supplementary material, videos S1 and S2). Each skin fragment was placed in a 50 ml falcon tube filled with seawater and brought to the boat; the tubes' seawater was

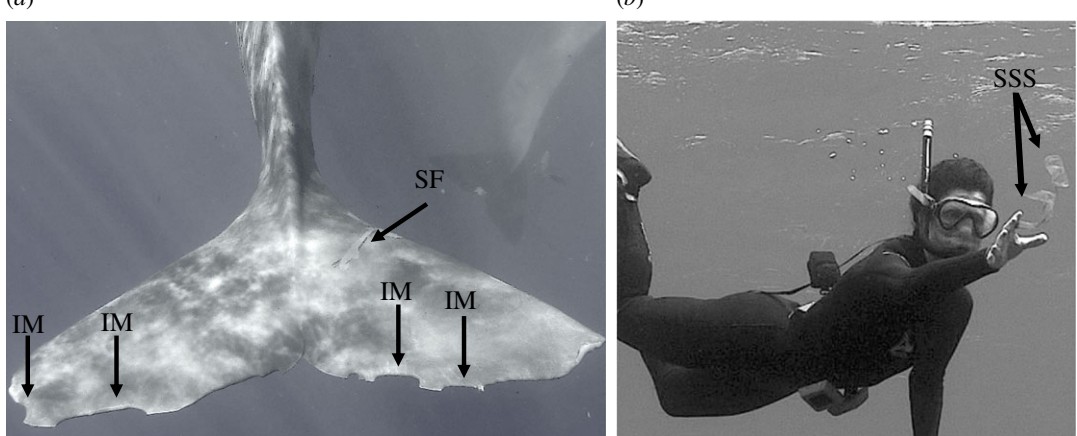

**Figure 2.** Examples of individual identification marks (*a*) and non-invasive collection of sperm whale skin samples (*b*). IM, identification marks; SF, skin fragment ready to be liberated; SSS, sloughed skin samples. (*a*), © V.S. and F.S., MMCO, Longitude 181. (*b*), © R.H., Label Bleu, MMCO, Longitude 181.

then removed and replaced with clean absolute ethanol, and skin fragments stored at room temperature and sent to Brest (France) under the CITES agreements FR1702900025-I and MU170203 (samples are listed in the electronic supplementary material, table S1).

## 2.3. DNA extractions

Genomic DNA was extracted from the skin samples with the NucleoSpin DNA RapidLyse® kit (Macherey-Nagel, Düren, Germany) following the manufacturer's recommendations, except for the lysis step, which was extended to 90 min. The quality and concentration of the extracted DNA were checked by agarose gel electrophoresis and by spectrophotometry using a NanoDrop 100 (Thermo Fisher Scientific, Waltham, USA). All DNA concentrations were standardized accordingly (10 ng $\mu l^{-1}$) for use in the polymerase chain reaction (PCR).

## 2.4. Analysis of mitochondrial DNA control region polymorphisms

A 774 bp fragment of the mtDNA control region (MCR) was amplified in presence of the DLP1.5 and DLP8G primers [55]. The ensuing PCR products were first purified using the NucleoSpin Gel and PCR Clean-up® kit (Macherey-Nagel) and then sequenced by a provider (GATC Biotech, Konstanz, Germany) in the presence of one of the two PCR primers. A fragment (638 bp) of the MCR was sequenced for all the sampled individuals, since this fragment included the hypervariable region of the D-Loop, which has been shown to contain polymorphic sites in sperm whales [56]. The sequences were manually edited and aligned in GENEIOUS PRO v.7.1 (Biomatters Ltd, Auckland, New Zealand). For each individual, the sequencing was performed in both directions for at least one DNA extract (see §2.5.1., 'Definition and quality control of individual-specific genotypes'). This fragment overlapped partially (602 bp in common) with those sperm whale MCR sequences determined by Alexander *et al.* [53] and fully with the data from Morin *et al.* [57]. We therefore constructed a new dataset that included all these sequences in order to permit a larger geographical analysis. The software DNASP v.5.10 [58] was used to identify the different haplotypes.

## 2.5. Analysis of nuclear microsatellite polymorphisms and estimation of kin relationships

Sixteen nuclear loci known to contain polymorphic microsatellite loci in different cetacean species were analysed: PPHO110, PPHO130, PPHO102, PPHO104, PPHO131, PPHO133 [59,60]; Ev1, Ev37, Ev94 [61]; GATA028, GATA417 [62]; 199–200, 417–418 [63]; GT023, GT211, GT575 [64] (see the electronic supplementary material, table S2 for more details on the PCR conditions). Amplified fragment sizes were determined on an Applied Biosystems 3130 Genetic Analyzer and analysed with the 'Microsatellite Plugin' of GENEIOUS PRO v.7.1.

### 2.5.1. Definition and quality control of individual-specific genotypes

In the field, when collected, each skin sample was assigned to one of the individuals identified. Skin samples were then anonymized with an alphanumeric code, and all steps of the genetic analysis were performed with anonymized skin samples. The software Cervus [65] was used to calculate the *pID* (probability of identity based on the allele frequencies, that is, the likelihood that two individuals randomly have the same genotype profile) allowing two divergent loci to take into account possible genotyping errors. Accordingly, similar genotypes were presumed to correspond to skin samples taken on different occasions from the same individual (all $pID < 2.11 \times 10^{-10}$). Once the similar genotypes were identified, *genetic individuals* were defined (see the electronic supplementary material, table S1). When only one skin sample was available for an individual, all the steps of the analysis were repeated (i.e. DNA extraction, MCR sequencing, microsatellite-containing loci genotyping). For each *genetic individual*, the genomic DNA was therefore extracted in at least two independent experiments and from different samples when available, and all the microsatellite loci were analysed from at least two independent DNA extracts. Seventeen individuals were sampled at least three times (electronic supplementary material, table S1). These different samples taken from the same individual let us perform replicate PCRs for all the nuclear loci and to estimate the genotyping errors linked to possible poor-quality DNA extracts [66,67]. We calculated an overall per-allele error rate of 2.2% (38 alleles incorrect among the 1708 scored) which was subsequently incorporated in the kinship analysis.

The match between *genetic individuals* and field-identified individuals was then tested for each skin sample. In the case of disagreement between the two identifications (i.e. a skin sample attributed to one individual in the field presented a genotype corresponding to another *genetic individual*), the first investigative step was to analyse the video recording of the skin sampling to determine whether the genetic individual could have been close to the sampling site, perhaps leading to a misidentification of the sloughed skin in the field.

### 2.5.2. Kinship analysis

The relatedness coefficient *r* estimates the degree of kin relationships between two individuals: an $r \approx 0.5$ reflects first-degree relationships (parent–offspring, full siblings); second-degree relationships (half-siblings, avuncular, grandparent–grandchild) have an $r \approx 0.25$, and for third-degree relationships (e.g. cousins), $r \approx 0.125$.

We used three different software tools to infer the relationships between all the genotyped sperm whale individuals. First, we estimated genetic relatedness between all dyads by calculating the relatedness estimator *r* by using the R package *Related* [68]. The relative power of differing relatedness estimators may vary depending on the dataset. We used a comparative function in *Related* to determine the most appropriate *r* estimator for our particular dataset from among the four offered (L&L [69], L&R [70], Q&G [71] and W [72]). We selected as estimators those with the highest correlation coefficient between observed and expected relatedness values.

Second, we used the software ML-RELATE [73] to calculate a relatedness coefficient based on the probabilities of sharing alleles identical by descent between two related individuals [74]. Based on this calculation, ML-RELATE can also determine the maximal likelihood familial relationship between two individuals (parent–offspring (PO), full sibling (FS), half-sibling (HS), unrelated). The test also computes a *p*-value to estimate the likelihood of this putative relationship (the smaller the *p*-value is, the better the putative relationship fits the data). When a specific relationship was suspected between two individuals, the validity of this *a priori* hypothesis was tested in ML-RELATE. We calculated two *p*-values: one for the more likely hypothesis versus the second most likely one, and a second *p*-value for the opposite situation. Tests were performed with 1000 simulations. The two *p*-values were then compared to determine which relationship was most attributable to the dyad (see below).

Third, the software Cervus 3.0.7 [65] was used to assign likely parentage, with all individuals as potential offspring, and all adult females as potential mothers. An individual was accepted as a likely parent with either a strict confidence level of 95% or a relaxed confidence level of 80%.

Based on the combined results of the above three analyses, all probable first- and second-degree kin relationships in the group were listed. All these relationships respected one of the following rules: (i) the software ML-RELATE indicates a familial relationship (PO, FS, or HS) between two individuals and its relatedness coefficients are coherent (at least two are >0.18 for a second-degree relationship, >0.40 for a first-degree relationship); (ii) in the case where ML-RELATE indicates a familial relationship (e.g. HS) inconsistent with the relatedness coefficients (at least two are <0.18 for a second-degree, <0.40 for a

first-degree relationship), the relationship was tested in ML-RELATE as described before. The relationship is retained if the value of the difference between the *p-values* of the two tests is < 0.025; and (iii) If ML-RELATE fails to indicate a familial relationship but at least two relatedness coefficients are >0.18, the dyad is tested in ML-RELATE for a second-degree relationship. The familial relationship is changed from unknown to half-sibling if the absolute value of the difference between the *p*-values of the two tests is ≤0.025.

# 3. Results

Between 2015 and 2019, 281 days of observations were conducted with about 1120 h of fieldwork and 250 h of video recorded. Irène's group was observed 86% of the time (in 242 of 281 field trips). Since 2011, we have always observed the same adult females ($n = 18$), of which just one disappeared, in 2015. There were 14 births, including three now-missing juveniles. Currently, the group is composed of 17 adult females and 11 juveniles.

A total of 92 skin samples were collected near the Mauritius Island between 2017 and 2019 (electronic supplementary material, table S1). These skin samples were assigned in the field to 27 distinct sperm whales: 24 adult females and juveniles belonging to Irène's group, plus two adult females listed in another social group (the 'Reshna group') and one unidentified adult female. Genomic DNA was successfully extracted from all samples, with yields ranging from 0.5 µg to 17 µg. Mitochondrial and nuclear loci were amplified from all DNA extracts, allowing for an analysis of the polymorphisms at 638 bp of the MCR as well as at 16 microsatellite loci (see the electronic supplementary material, table S3 for the number of alleles per locus). No major variations in the quality of the DNA extracted from the skin samples were noticed, contrasting with previous results [66,75]. This is most likely owing to the fact that the skin samples were taken immediately after their release from the whales' body. The 'identity analysis' performed in CERVUS allowed us to identify 27 genetic individuals, all having a *pID* < $2.11 \times 10^{-10}$ (electronic supplementary material, table S1). All samples but one each from Aiko (2017_03A) and Zoé (2018_51B) that had lower quality genotypes (too many missing alleles) were successfully matched to at least another genotype in the identity analysis. Because more than three other samples of Aiko and Zoé had been taken, those two bad-quality samples were removed from the analysis. All fuzzy matching genotypes identified in the identity analysis were analysed in detail: in all cases, they arose from the presence of a missing (null) allele in one of the genotypes of the same individual and were therefore considered a correct match. Mitochondrial haplotypes were identical between all samples of the same individual. Finally, the 27 genetic individuals identified based on similar genotypes (electronic supplementary material, table S1) matched up with the 27 field-identified sperm whales. Only four skin samples had to be reassigned to another sperm whale after their genotyping, i.e. less than 5% of field misidentification when collecting samples. For all these four cases, a careful *posteriori* examination of the video recording was sufficient to explain the misidentification made.

## 3.1. Kinship relations in Irène's sperm whale group

### 3.1.1. All members of Irène's group but one presented the same mitochondrial DNA haplotype

Irène's sperm whale group included 28 members, all of them clearly identified on the basis of individual external marks [54] (figure 2). Sloughed skins were sampled from 24 of them during this study (adult females, $n = 15$; young females, $n = 2$; young males, $n = 7$; electronic supplementary material, table S1). All of the 24 Irène's group members, except one (Claire), presented the same haplotype for this 638 bp fragment, named SW_M1. This haplotype corresponds to the haplotype C.001.002 as defined by Alexander *et al.* [53], a haplotype considered rare in the western Indian Ocean.

Claire harboured another haplotype, named SW_MCK1, differing by two transitions (G− > A) from SW_M1. This SW_MCK1 corresponds to haplotype N.001.001 of Alexander *et al.* [53], which is highly represented in the Seychelles but has a minor prevalence in the rest of the Indian Ocean.

### 3.1.2. Microsatellite polymorphisms highlighted numerous close kin relationships in Irène's group

We retained the Wang $r_W$ and the Lynch Li $r_L$ estimators for the relatedness coefficient (*r*) calculation (electronic supplementary material, figure S1) and compared them with the $r_K$ calculated by ML-RELATE [73]. The relatedness values for all dyads can be found in the electronic supplementary material, table S4. Parentage analysis was also implemented in CERVUS on all the dyads.

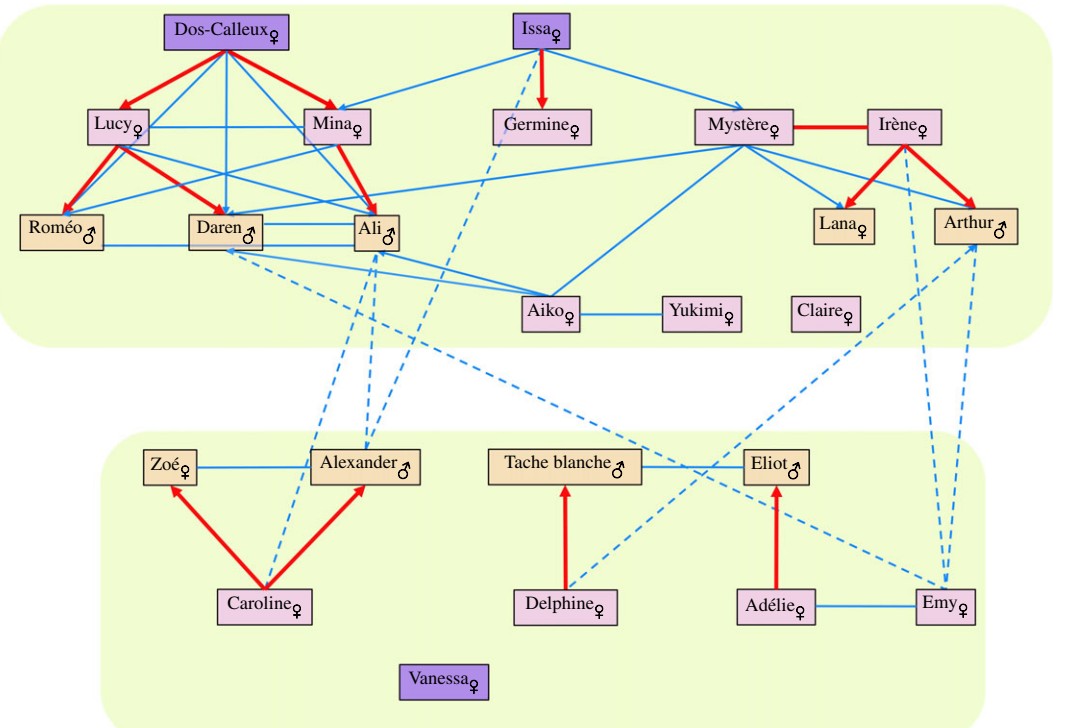

**Figure 3.** Schematic representation of the kin relationships between all the members of Irène's sperm whale clan. All first-degree (red lines) and second-degree (solid and dotted blue lines) relationships between the different sperm whales are represented. The name and sex are indicated for each individual. Adult sperm whales are indicated in purple (dark for older individuals, as estimated in the field, and light purple for others), with the young sperm whales in yellow. The two green boxes represent the two social subgroups identified; second-degree relationships are distinguished by a solid line for those in the same subgroup, and by a dotted line when they link sperm whales of the two subgroups. This diagram was constructed to be consistent with the analyses conducted. Although we performed different analyses that produced similar results, uncertainty exists in the relatedness estimate calculations, which might influence some of these relationships.

Sixteen first-degree relationships (parent–offspring and full siblings) were inferred from the four calculations (electronic supplementary material, table S5). Among them, three shifted to second-degree relationships (see the electronic supplementary material, table S5 for a detailed explanation). Four of the first-degree relationships each involved two adult females. Specifically: (i) Mystère and Irène shared such a relationship, which may reflect either a full–sibling one (which we retained as more probable) or a mother–offspring one; (ii) Germine was a juvenile in 2013 (her teeth were still emerging), so Issa has to be her mother; and (iii) the case was more complex to decipher for Dos-Calleux, Lucy, and Mina—but sperm whale adult females rarely give birth after 40 years of age [13], which makes it highly unlikely that the mother of Dos-Calleux could also be that of a young sperm whale born in 2018 (e.g. Daren and Ali). The most probable relationship is therefore the one conveyed in figure 3, where Dos-Calleux is the mother of both Mina and Lucy. Overall, 34 second-degree relationships were identified (electronic supplementary material, tables S5, S6). The average $r$ for Irène's group was $r = 0.048$ ($r = 0.035$ for females only; electronic supplementary material, table S7).

Based on this list of relationships, a diagram representing all the first- and second-degree links was drawn (figure 3). This diagram emphasizes the high density of relationships connecting almost all members of Irène's group. Among the adult females, Claire and Vanessa shared no first- or second-degree relationships with other members of the group, while Yukimi shared just one second-degree relationship with an adult female. In addition, Claire presented a different mitochondrial haplotype, thus confirming her stark genetic difference from the rest of the group's members.

### 3.1.3. Genetic relationships in the two subgroups of Irène's group

A total of 81 days of fieldwork were completed in 2019. Instead of Irène's group (observed only 28 times in 2019), two subgroups were regularly observed: the subgroup I included 15 sperm whales (observed

27 times in 2019), and the subgroup II had nine sperm whales (observed 12 times in 2019). The electronic supplementary material, table S1 lists the members of each subgroup. Figure 3 shows that no first-degree relationships could be found between sperm whales of the different subgroups. Members of subgroup I presented a high density of first- ($n = 9$) and second-degree ($n = 17$) relationships, reflected by an average $r = 0.08$ ($r = 0.072$ for females only; electronic supplementary material, table S7). The subgroup II consisted of 5 adult females and 4 young sperm whales, being mainly composed of mother–offspring pairs (4 first-degree relationships, and 3 second-degree relationships). This composition explained why the average $r$ for the females of this subgroup II ($r = 0.07$) was lower than that of the whole subgroup II, $r = 0.11$ (electronic supplementary material, table S7).

Only 7 second-degree relationships were found between members of the two subgroups, and just one involved two adult females.

## 3.2. Genetic relationships between Irène's group members and other sperm whales sampled at Mauritius island

Two adult females belonging to another social group (Reshna's group, F. Sarano and V. Sarano 2018, unpublished observations) were sampled in the Mauritius Island. They presented the SW_M1 haplotype, and some second-degree relationships with members of Irène's group (electronic supplementary material, table S8), possibly suggesting that they belonged to social groups close to Irène's group.

# 4. Discussion

We studied a stable social group of sperm whales, named Irène's group, composed of 28 sperm whales (17 adult females, 11 juveniles) observed together regularly (at least 242 times since 2015) off Mauritius Island which most likely corresponds to a social unit, as defined by previous studies [4,38]. Altogether, and after successful comparisons between *genetic individuals* identified via anonymized samples and from field-identified individuals, 24 sperm whales of Irène's group were represented in the samples (electronic supplementary material, table S1), i.e. 24 out of 28 individuals. Irène's group is large for a sperm whale social unit, but not unprecedented [38]. This may be owing to geographical specificities (in the Indian Ocean, sperm whales are understudied), and it may well be the reason for the ongoing scission of the group that have been observed (see below).

## 4.1. Irène's sperm whale group is matrilineal and shows extensive allomaternal care

Matrilineal groups are thought to be composed of an older female (the maternal ancestor) and all her offspring [48]. For such a long-lived species, several generations are likely to be present in sperm whale social groups, while the maternal ancestor is no longer present (provided that its presence is not a requisite for the group stability, see below). If sperm whale social groups are strictly matrilineal, all members of the same group should therefore *a priori* present the same mtDNA because of its maternal transmission. Except for that one adult female named Claire—which will be discussed below—all of the sperm whales in Irène's group share the same MCR haplotype, SW_M1, corresponding to the haplotype C.001.002 of Alexander *et al.* [53]. This C.001.002 has a worldwide representation, but it is a minor in the Indian Ocean where it can be found mainly in Aldabras (west of the Seychelles [53]). Thus, it is extremely unlikely that this mtDNA homogeneity originated simply from a geographical haplotype prevalence. However, the skin sampling needs to be locally extended to other social groups, in order to estimate the actual frequency of SW_M1 in the local population of sperm whales. The matrilineality of Irène's group is nevertheless strongly confirmed here by our results. This agrees with the findings of Konrad *et al.* [75] for the Atlantic Ocean, though they identified only two mtDNA haplotypes in 12 different social groups, which was low to conclude about matrilineality.

In Irène's group, only one adult female, Claire, possesses a different haplotype, SW_MCK1, which differs by two mutations from SW_M1. Moreover, Claire is not closely related to any member of the group, and she also has the lower average relatedness coefficient $r$ (see below) among all members of the group ($r = -0.015$). It is therefore likely that Claire was not born within Irène's social group. Two other females, Vanessa and Yukimi, despite sharing the same haplotype as the rest of the group, have no or few second-degree relationships. Konrad *et al.* [75] showed that some individuals may present no clear genetic relationships with other members of their social groups. Transfers of sperm whales

between social units were observed before but have not been genetically characterized [38]. The intriguing case of Claire, presenting both a different mtDNA haplotype and a low relatedness coefficient with the other members of the group, confirms the likelihood of such an event, and demonstrates that this kind of group 'adoption' is not necessarily based on kin relationships.

In contrast with mtDNA, nuclear DNA does not necessarily reveal high relatedness between all individuals of a matrilineal group. The relatedness coefficient $r$ underwent a twofold decrease with each increasing degree of relationships (i.e. $r \approx 0.5$ for first-degree relationships, $r \approx 0.25$ for second-degree ones, $r \approx 0.125$ for third-degree, …). The plurality of breeding females in the group and the *a priori* high diversity of the fathers added to their short breeding tenure can also increase the nuclear diversity in a matrilineal group [76]. We found a high density of first-, second- and third-degree relationships between all members of Irène's group (figure 3 and electronic supplementary material, tables S5 and S6, for first- and second-degrees), but obtained an overall $r = 0.048$ between all members of Irène's group (electronic supplementary material, table S7), although this value would require the inclusion of more genotypes, from other social groups, in the dataset to be significant.

Alloparental care and alloparental nursing were commonly observed in Irène's group, confirming previous observations [67,77]. For instance, an adult female, Emy, was identified in the field as the mother of Alexander, a juvenile male, but no kin relationship connected them. Our study showed that Caroline was in fact the genetic mother of Alexander (figure 3, electronic supplementary material, table S5), although both were rarely observed associated (on eight filmed Alexander-nursing observations, seven implied Emy and only one Caroline; V. Sarano and F. Sarano 2015–2020, unpublished observations). Many other cases of allonursing were also witnessed. Moreover, an adult female, Germine, was observed taking care of all juveniles at different times, which could represent a case of an individual specialized in nursing.

## 4.2. Kin relationships, and in particular first-degree ones, seem to shape the ongoing scission of Irène's group

From conservation and evolutionary perspectives, understanding the dynamic of social group functioning is of major importance because it can affect individual fitness, life history, genetic diversity and cultural transmission, among other parameters. In addition to the transfer of individuals between social units, the long-term monitoring of Irène's sperm whale group also highlighted a likely ongoing scission of the group. Two subgroups of stable composition (see the electronic supplementary material, table S1) have been commonly observed over the last few years instead of the full group that had been regularly observed since 2013. Scission in a sperm whale social group is a very rare event [75]. It has been previously described just once by Christal *et al.* [38], though with no information about the kinship pattern of the scission. The presence of a matriarch (or the common ancestor) is thought to promote group cohesion [78,79] and its disappearance may therefore be considered as a putative factor driving scission [80]. Here, only one adult female identified in Irène's group before 2015 was not seen after this time, and three calves disappeared, respectively, in 2011, 2015 and 2017 [54]. None of these disappearances can confidently explain the group scission that seems to be occurring. The most likely explanation remains group size, because it plays a major role in social group stability in mammals (i.e. [81,82]): the preservation of an optimal size could represent a probable explanation for the scission in Irène's group. Eleven juveniles were born and have survived since 2011 in the group (electronic supplementary material, table S1), leading to possible overcrowding, for instance, with respect to food resources, finally causing the group to split.

The kin relationships pattern of this Irène's group scission can be clearly analysed (figure 3). An outstanding point is that no first-degree relationships were broken: all mother–calf dyads belong to the same subgroup. It is worth noting that both the genetic mother and the 'nurse' of Alexander (respectively Caroline and Emy) belong, with Alexander, to the same subgroup. The subgroup I groups individuals presented several second- and third-degree relationships, which is revealed by an average $r = 0.08$, almost twice that of Irène's group ($r = 0.048$); hence, this subgroup I is mainly composed of closely related individuals. The composition of subgroup II looks quite different with regard to kin relationships: four mother–child pairs, with no strong relationships between the adult females (figure 3). There were less second-degree relationships between adult females of subgroup II than between members of the two subgroups (compare the solid and dotted blue lines of figure 3). This is reflected by a strong difference in the $r$ for the females of subgroup II ($r = 0.07$) versus the whole subgroup ($r = 0.11$). All these observations (no first-degree relationships broken and the average

kinship in each subgroup increased) are in agreement with the results of Van Horn *et al.* [83], who observed that matrilineal groups often split along matrilineal lines and that average levels of kinship between group members increase. Here, it appears to be the case as well, although membership to a subgroup is not exclusively dependent on genetic closeness (see for instance Claire and Yukimi, both members of subgroup I).

## 4.3. The influence of complex sperm whale cultural societies on the species evolution: current understanding and perspectives

The multilevel sperm whale society is certainly one of the most complex societies in mammals [4]. The behavioural differences between sexes are extreme, from highly social philopatric females to 'long-distance running' males. We have highlighted here the strict matrilineality of a well-studied social group. Similar conclusions were drawn by Konrad *et al.* [75] about Caribbean sperm whales. Further studies are required now to generalize those results to other ocean basins. The female philopatry and male-mediated gene flow, resulting from the social organization of the species, has shaped its evolution, as revealed by several phylogeographic studies [53,57,84,85]. Still, Morin *et al.* [57] demonstrated that an historically small population of sperm whales was another major factor contributing to present low genetic diversity of the species.

The influence of cultural traits on sperm whale evolution is still under much questioning. The main cultural trait in sperm whales is the transmission of codas, whose specificities define vocal clans grouping different stable social groups, like the one we studied here at Mauritius Island. These vocal clans, called cultural groups [4], also share other behavioural traits, namely those concerning foraging and socializing [86]. Gene–culture coevolution requires *a priori* both cultural trait(s) that are socially transmitted and a barrier to gene flow between cultural groups. However, in humpback whales, where gene–culture coevolution is obvious, because the vertical cultural transmission of migration routes shapes the geographical distribution of mitochondrial genes, barriers to gene flow are not always obvious [24]. In sperm whales, social clans lead to different distributions of mtDNA haplotypes, and phylogeographic studies have highlighted that the mtDNA distribution was influenced more by social group membership than by geographical distribution [53,87]. Rendell *et al.* [88] uncovered some correlations between mtDNA divergences and coda dialect in Pacific Ocean sperm whales. However, no simple link was found (like, for instance, one mtDNA haplotype associated with a particular coda), thus providing no clear support for the hypothesis of a vertical co-transmission of mtDNA and of coda learning.

If the sperm whale is without question one of the more cultural species of the marine realm, deciphering the possible influence of culture upon its evolution still requires further understanding, some of which has been highlighted by our study. Fundamental questions nonetheless remain on the matrilineality of cultural units in sperm whales [4], and on the selection of coding genes of the nuclear and mitochondrial genomes possibly linked to the cultural trait that are socially transmitted [57].

Ethics. Permission to conduct the *Maubydick* project, including the taking of sloughed skin fragments, was granted by the Department for Continental Shelf, Maritime Zones Administration and Exploration of the Mauritius Prime Minister Office, on the 21 February 2017. Skin samples were sent to Brest (France) under the CITES agreement FR1702900025-I.

Data accessibility. DNA sequences are deposited in the Genbank under references MK907143-45; MK907149-58; MK907160-62; MK907164-69; MK907171 and MK907173-76. Microsatellite data are accessible at https://doi.org/10.5061/dryad.bcc2fqzbk [89].

Authors' contribution. Both F.S. and J.L.J. designed the study. F.S., O.A., F.D., H.G. and J.L.J. contributed variously to the conception of the project. F.S., V.S., R.H., A.P. and H.V. performed the field experiments and identification of individual sperm whales. J.G., J.L.J. and A.M.G.S. conducted the genetic analysis (laboratory procedures). J.G. and J.L.J. analysed and interpreted the genetic data J.L.J. and J.G. wrote the manuscript. F.S., V.S., B.M., A.M.G.C., F.D., O.A. and H.G. critically revised the manuscript.

Competing interests. The authors declare no competing interests.

Funding. Genetic analyses were granted by Lush-France (Paris, France) and by the University of Brest (France). Fieldwork, video recording and data analysis were granted by the NGOs Longitude 181 (Valence, France) and 'Un Océan de Vie' (Marseille, France), by the 'Label Bleu' Film Production (Marseille, France), by the Foundation 'Nature et Découvertes' (http://www.fondation-natureetdecouvertes.com) and by Teria (Vitry sur Seine, France).

Acknowledgements. Mauritian public authorities greatly helped the *Maubydick* project, in particular the Mauritian Prime Minister Office, the Marine Continental Shelf Exploration and Administration (MCSEA; Dr Réza Badal and his team), the Albion Fisheries Research Center (AFRC; Chief Scientific officer Mr Satish Kadhun), the Mauritius Film Development Corporation (MFDC; Mr Sachin Jootun and Miss Eliana Timol) and the Tourism Authority (TA; Miss

Khoudijah Boodoo, Director). We thank Navin Boodhonee and the Blue Water Diving Center (Trou aux Biches, Mauritius) for their valuable participation in the fieldwork. We also thank Carole Decker (Brest, France) for her help with the laboratory experiments. This manuscript benefitted from constructive comments provided by two anonymous reviewers.

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
