## [Peer Review File · Royal Society Open Science]

Kin relationships in cultural species of the marine realm: case study of a matrilineal social group of sperm whales off Mauritius Island, Indian Ocean

Francois Sarano, Justine Girardet, Véronique Sarano, Hugues Vitry, Axel
Preud'homme, René Heuzey, Ana Maria Garcia Cegarra, Bénédicte Madon,
Fabienne Delfour, Hervé Glotin, Olivier Adam and Jean-Luc Jung

Article citation details

R. Soc. open sci. **8**: 201794.
<http://dx.doi.org/10.1098/rsos.201794>

Review timeline

Original submission: 24 January 2020
1st revised submission: 6 October 2020
2nd revised submission: 2 December 2020
3rd revised submission: 7 January 2021
Final acceptance: 11 January 2021

Note: Reports are unedited and appear as
submitted by the referee. The review history
appears in chronological order.

Review History

RSOS-200086.R0 (Original submission)

Review form: Reviewer 1

Is the manuscript scientifically sound in its present form?

No

Are the interpretations and conclusions justified by the results?

No

Is the language acceptable?

No

Do you have any ethical concerns with this paper?

No

Have you any concerns about statistical analyses in this paper?

Yes

Recommendation?

Reject

Comments to the Author(s)

This manuscript describes work to assess the patterns of relatedness within a sperm whale social unit in waters off Mauritius Island. For the most part the work seems good and the results interesting. However, I have a fair number of concerns regarding both the theoretical context of the paper, as well as with the application and interpretation of some of the analyses. I have outlined these below. I have divided my comments into larger-scale comments, and smaller-scale edits to the manuscript.

Larger-Scale Comments

1. As a whole, the entire section describing different aspects of social structure, and the relative strengths and weaknesses of group living could use stronger theoretical foundations. Some of these are listed below, but this whole section could be strengthened.

a. Lines 86-88: This idea needs explanation. The authors should clarify what they mean by, and explain the underlying theory, behind the statement “Low average relatedness...could be correlated in particular to high individual social relationships disparities..” I don’t follow this at all.

b. Lines 91-92: The statement “Group size and relatedness are also correlated, in a species-specific manner” needs much more explanation/justification.

2. The context of what is known about sperm whale social groups does not seem to be properly represented, and therefore this whole section could use revision. For example Lines 140-153 give the impression that this is not well known, and that, for some unknown reason, different studies produce different results. However, this is not necessarily true, and is certainly inaccurate. Instead, we know that the social structure of sperm whale female units differs fairly substantially in different regions of the world (e.g., Whitehead & Kahn (1992) *Can. J. Zool.* 70: 2145-2149, Whitehead (2003) *Sperm Whales: Social Evolution in the Ocean*). Therefore, we would expect to obtain different results due to the differences in underlying biology in different areas, rather than due to some sort of analytical discrepancy.

3. The importance of underwater observations. The authors suggest that this aspect of their work is quite important, but I don’t ever see this justified satisfactorily. Why, or under what conditions, are underwater observations better? The logistics seem difficult to me. If a group of 28 sperm whales swims by, on how many occasions can a snorkeler get identification photographs of all 28? Collecting data from the water seems fine, but is not adequately justified here. The authors should thoroughly explain why they are taking this approach rather than vessels.

4. Not enough information is provided on the genetic mismatches/genotyping errors.

a. Lines 300-302: How many alleles could differ between two genotypes and have them still be considered “similar”. This should be explicit, and well-justified.

b. Lines 302-304: Once “genetic individuals” were identified, which genotype was used as representative of that individual, and how was that decided? This could have big implications for the relatedness analyses.

c. What are the units for this error rate. Is this per amplification, per allele, per locus, or per genotype? These all mean very different things, but it is not clear what is being referred to here, or how this number was obtained.

5. Interpretation of relatedness analyses. All relatedness estimates are based on comparison to a “reference group”, which is usually all of the genotypes of the study (unless otherwise specified). All relatedness estimators are standardized so that the average relatedness within a sample set is 0 (or approximately so). Because this group consisted of genotypes from one social unit, it is not appropriate for estimating the average relatedness within that unit (it will be about zero). To do this, many more genotypes, from other social groups, would be needed. Then it would be possible to say something about the relatedness of individuals within this group relative to the population as a whole. However, since all but two genotypes are from individuals within this group, it is not appropriate to use these data to estimate the average relatedness within the group. Therefore, the estimate obtained (average $r = 0.048$) does not mean anything, because the appropriate reference group has not been sampled. Therefore, all aspects of this should be removed from the Results and Discussion. It is still appropriate to use these data to identify specific relationships among pairs of individuals, but even here their estimated r -values will be lower than what they truly are, again because there is not an appropriate reference group for comparison.

Smaller-Scale Edits

1. Line 49: The statement that “Genotypes matched field identifications” is not accurate. There were some cases where this was not true, at least initially, and also such a statement is not possible (a genotype can’t match a field identification). I would just remove this sentence.

2. Line 75: Should be *Physeter macrocephalus* (!)

3. Lines 96-97: The statement “...most individuals are undistinguishable” requires more justification. I know of many cetacean species for which every individual can be identified, and therefore this statement does not seem accurate.

4. Lines 174-175: “..attributed with enough confidence...” what does this mean? This statement requires much more explanation.

5. Line 99-100: I don’t see how the previous sentences, regarding photo-ID and genetic sampling, lead to interest in social groups. This tie should be more clear.

6. Line 103: What is meant by “of first importance”?

7. Line 105: “nowadays” seems too informal for a scientific publication.

8. Line 120: What is meant by “outsize body”? Perhaps different wording.

9. Line 133-135: This sentence should be removed. The fact that no migrations were known, until recently, does not add to the paper.

10. Line 248: CITES requires both an import and an export permit, yet this is just one number. What is the other one?

11. Line 251: I thought that biopsy samples were not collected (that all were sloughed skin).
12. Line 258: What concentration was the DNA samples standardized to?
13. Lines 265-272: More details regarding microsatellite amplification are required: annealing temperatures, were they amplified singly or in multiplex reactions, primer concentrations, fluorescent labels, etc. This could be a table in the Supplementary Material. Moreover, how were they size-separated and visualized? What sort of equipment?
14. Lines 546-549: This statement is not true. Konrad et al. (ref. 60) used genetic data to show that some individuals in social groups did not have close genetic ties therein, much like “Claire” here.
15. Line 568: Should be Clutton-Brock.
16. Lines 568-579: This does not seem relevant to this study.
17. Lines 603-604: This sentence requires justification. Why is understanding this of major importance? To who?
18. Line 663-666: Should cite ref. 60 again here, as Konrad et al. characterized the matrilineal nature of sperm whale social groups in a different ocean basin.
19. Lines 697-705: Not sure this is immediately relevant to this study.

Review form: Reviewer 2

Is the manuscript scientifically sound in its present form?

No

Are the interpretations and conclusions justified by the results?

No

Is the language acceptable?

Yes

Do you have any ethical concerns with this paper?

No

Have you any concerns about statistical analyses in this paper?

No

Recommendation?

Major revision is needed (please make suggestions in comments)

Comments to the Author(s)

Sarano et al. present a high-resolution study of kin relationships within a social group of sperm whales off the coast of Mauritius. While limited in scope by the small sample size and geographic range, the paper represents an important contribution toward understanding links between social structure and genetic structure in a social cetacean, the sperm whale.

The data collection and analysis methods are sound, but the authors over-interpret their results in a few places without sufficient support for their conclusions. I have made several comments in the attached document (Appendix A), highlighting places throughout the manuscript where the authors can rephrase or shift their interpretation of the results to more accurately reflect the data presented. I also make recommendations for additional background information that should be added to the introduction.

Decision letter (RSOS-200086.R0)

04-Mar-2020

Dear Dr Jung:

Manuscript ID RSOS-200086 entitled "Kin relationships in cultural species of the marine realm: case study of a matrilineal social group of sperm whales off Mauritius Island, Indian Ocean" which you submitted to Royal Society Open Science, has been reviewed. The comments from reviewers are included at the bottom of this letter.

In view of the criticisms of the reviewers, the manuscript has been rejected in its current form. However, a new manuscript may be submitted which takes into consideration these comments.

Please note that resubmitting your manuscript does not guarantee eventual acceptance, and that your resubmission will be subject to peer review before a decision is made.

Your resubmitted manuscript should be submitted by 01-Sep-2020. If you are unable to submit by this date please contact the Editorial Office.

on behalf of Professor Michael Bruford (Associate Editor) and Pete Smith (Subject Editor)
openscience@royalsociety.org

Reviewers' Comments to Author:

Reviewer: 1

Comments to the Author(s)

This manuscript describes work to assess the patterns of relatedness within a sperm whale social unit in waters off Mauritius Island. For the most part the work seems good and the results interesting. However, I have a fair number of concerns regarding both the theoretical context of the paper, as well as with the application and interpretation of some of the analyses. I have outlined these below. I have divided my comments into larger-scale comments, and smaller-scale edits to the manuscript.

Larger-Scale Comments

1. As a whole, the entire section describing different aspects of social structure, and the relative strengths and weaknesses of group living could use stronger theoretical foundations. Some of these are listed below, but this whole section could be strengthened.
 - a. Lines 86-88: This idea needs explanation. The authors should clarify what they mean by, and explain the underlying theory, behind the statement "Low average relatedness...could be correlated in particular to high individual social relationships disparities.." I don't follow this at all.
 - b. Lines 91-92: The statement "Group size and relatedness are also correlated, in a species-specific manner" needs much more explanation/justification.
2. The context of what is known about sperm whale social groups does not seem to be properly represented, and therefore this whole section could use revision. For example Lines 140-153 give the impression that this is not well known, and that, for some unknown reason, different studies produce different results. However, this is not necessarily true, and is certainly inaccurate. Instead, we know that the social structure of sperm whale female units differs fairly substantially in different regions of the world (e.g., Whitehead & Kahn (1992) *Can. J. Zool.* 70: 2145-2149, Whitehead (2003) *Sperm Whales: Social Evolution in the Ocean*). Therefore, we would expect to obtain different results due to the differences in underlying biology in different areas, rather than due to some sort of analytical discrepancy.
3. The importance of underwater observations. The authors suggest that this aspect of their work is quite important, but I don't ever see this justified satisfactorily. Why, or under what conditions, are underwater observations better? The logistics seem difficult to me. If a group of 28 sperm whales swims by, on how many occasions can a snorkeler get identification photographs of all 28? Collecting data from the water seems fine, but is not adequately justified here. The authors should thoroughly explain why they are taking this approach rather than vessels.
4. Not enough information is provided on the genetic mismatches/genotyping errors.
 - a. Lines 300-302: How many alleles could differ between two genotypes and have them still be considered "similar". This should be explicit, and well-justified.
 - b. Lines 302-304: Once "genetic individuals" were identified, which genotype was used as representative of that individual, and how was that decided? This could have big implications for the relatedness analyses.
 - c. What are the units for this error rate. Is this per amplification, per allele, per locus, or per genotype? These all mean very different things, but it is not clear what is being referred to here, or how this number was obtained.
5. Interpretation of relatedness analyses. All relatedness estimates are based on comparison to a "reference group", which is usually all of the genotypes of the study (unless otherwise specified).

All relatedness estimators are standardized so that the average relatedness within a sample set is 0 (or approximately so). Because this group consisted of genotypes from one social unit, it is not appropriate for estimating the average relatedness within that unit (it will be about zero). To do this, many more genotypes, from other social groups, would be needed. Then it would be possible to say something about the relatedness of individuals within this group relative to the population as a whole. However, since all but two genotypes are from individuals within this group, it is not appropriate to use these data to estimate the average relatedness within the group. Therefore, the estimate obtained (average $r = 0.048$) does not mean anything, because the appropriate reference group has not been sampled. Therefore, all aspects of this should be removed from the Results and Discussion. It is still appropriate to use these data to identify specific relationships among pairs of individuals, but even here their estimated r -values will be lower than what they truly are, again because there is not an appropriate reference group for comparison.

Smaller-Scale Edits

1. Line 49: The statement that “Genotypes matched field identifications” is not accurate. There were some cases where this was not true, at least initially, and also such a statement is not possible (a genotype can’t match a field identification). I would just remove this sentence.
2. Line 75: Should be *Physeter macrocephalus* (!)
3. Lines 96-97: The statement “...most individuals are undistinguishable” requires more justification. I know of many cetacean species for which every individual can be identified, and therefore this statement does not seem accurate.
4. Lines 174-175: “..attributed with enough confidence...” what does this mean? This statement requires much more explanation.
5. Line 99-100: I don’t see how the previous sentences, regarding photo-ID and genetic sampling, lead to interest in social groups. This tie should be more clear.
6. Line 103: What is meant by “of first importance”?
7. Line 105: “nowadays” seems too informal for a scientific publication.
8. Line 120: What is meant by “outsize body”? Perhaps different wording.
9. Line 133-135: This sentence should be removed. The fact that no migrations were known, until recently, does not add to the paper.
10. Line 248: CITES requires both an import and an export permit, yet this is just one number. What is the other one?
11. Line 251: I thought that biopsy samples were not collected (that all were sloughed skin).
12. Line 258: What concentration was the DNA samples standardized to?
13. Lines 265-272: More details regarding microsatellite amplification are required: annealing temperatures, were they amplified singly or in multiplex reactions, primer concentrations, fluorescent labels, etc. This could be a table in the Supplementary Material. Moreover, how were they size-separated and visualized? What sort of equipment?

14. Lines 546-549: This statement is not true. Konrad et al. (ref. 60) used genetic data to show that some individuals in social groups did not have close genetic ties therein, much like “Claire” here.

15. Line 568: Should be Clutton-Brock.

16. Lines 568-579: This does not seem relevant to this study.

17. Lines 603-604: This sentence requires justification. Why is understanding this of major importance? To who?

18. Line 663-666: Should cite ref. 60 again here, as Konrad et al. characterized the matrilineal nature of sperm whale social groups in a different ocean basin.

19. Lines 697-705: Not sure this is immediately relevant to this study.

Reviewer: 2

Comments to the Author(s)

Sarano et al. present a high-resolution study of kin relationships within a social group of sperm whales off the coast of Mauritius. While limited in scope by the small sample size and geographic range, the paper represents an important contribution toward understanding links between social structure and genetic structure in a social cetacean, the sperm whale.

The data collection and analysis methods are sound, but the authors over-interpret their results in a few places without sufficient support for their conclusions. I have made several comments in the attached document (Sarano et al. comments.pdf), highlighting places throughout the manuscript where the authors can rephrase or shift their interpretation of the results to more accurately reflect the data presented. I also make recommendations for additional background information that should be added to the introduction.

Author's Response to Decision Letter for (RSOS-200086.R0)

See Appendix B.

RSOS-201794.R0

Review form: Reviewer 1

Is the manuscript scientifically sound in its present form?

Yes

Are the interpretations and conclusions justified by the results?

Yes

Is the language acceptable?

No

Do you have any ethical concerns with this paper?

No

Have you any concerns about statistical analyses in this paper?

No

Recommendation?

Major revision is needed (please make suggestions in comments)

Comments to the Author(s)

The manuscript is improved from the first draft; however, there is still quite a bit of room for improvement, although perhaps it is more cosmetic this time. I found in some parts, particularly the Abstract, but also elsewhere, the work was over-sold and the wording should be toned down. Additionally, I think the context of our knowledge of sperm whale social structure is not represented accurately in the Introduction, which provides a false context for this paper. Other than those, I have several more specific suggestions to improve the paper, as described below.

1. Line 48-49: I think it is overselling it to call the sampling technique “innovative” (it is just collecting skin that is seen floating in the water after the animals pass, after all). I suggest changing this to “using underwater observations, individual-specific identification, and genetic analyses based on mitochondrial sequencing and microsatellite profiling.”
2. Lines 53-55: This statement is incorrect. Konrad et al. (2018) did the same thing for multiple groups in their paper (some of which were completely sampled).
3. Line 57: I think the word “adoption” here could be misleading. Technically, this would mean that a mother nurses and raises a calf that is not her own (which has been reported in whales). But it is not clear that that happened here, and it would be easy for readers to get confused. I would use clearer wording, perhaps “We highlight a likely case of an unrelated female being integrated into a social unit, presenting a mtDNA...”
4. Lines 81-83: I think “strongly dependent on kin relationships” is an over-statement. For some of these species it is definitely true, but I think the jury is still out on the importance of relatedness for some of them (e.g., long-finned pilot whales).
5. Line 91: It would be helpful if the authors define what is meant by a “fission-fusion model” so that all of the readers are on the same page.
6. Line 104: This isn't quite right. These aren't “genetically distinct” populations. It is true that maternally-directed site-fidelity in baleen whales leads to distinct structure of mitochondrial haplotypes between whales in different feeding grounds. But often whales from different feeding grounds all breed together in the winter. Thus, the feeding groups represent seasonal population structure, but certainly not “genetically distinct populations”.
7. Lines 171-172: The statement that “all samples used in these studies were biopsies...” is not true. Many are based on sloughed skin.
8. Lines 173-177: This doesn't provide an accurate picture of the studies underlying our current knowledge of sperm whale social structure. It is very hand-wavy and inaccurate. This may be

true for some populations, but not for others. Painting these studies in this way sets them up as a straw-man, presumably to make this current study look better. This does not seem appropriate.

9. Lines 222-224: I think is is over-selling it. Although perhaps true for this particular geographical region, social structure in sperm whales has been well-studied other places. This sentence also does not add anything to the paper.

10. Lines 292-306: It is a bit weird that some description of the mitochondrial sequencing and microsatellite genotyping are provided here, but then other aspects of these procedures are described separately, under their own subheadings. It would be clearer if the text from here was moved to its appropriate subheading, so that all of the information describing each procedure was kept together. Also on line 292 it states that 774 bp of the control region was amplified, but on line 310 it says 638. Why the difference?

11. Line 356: More information should be provided about how this error rate of 2.2% was calculated. Is it per allele, per locus, per genotype? Exactly how many discrepancies were identified? The authors should be very explicit in this explanation. I had this comment last time, and the authors address it in their response, but it should be added to the manuscript.

12. Lines 512-523: For adult females that were already adult when the study started, and that showed parent-offspring relatedness with each other, how did they authors decide who was the mother and who was the offspring (since all were adult upon the initiation of the study). This should be clearly explained.

13. Figure 2: I think the diagram is an intuitive and informative way to visualize the putative relationships. However, my concern is that they could be interpreted as "fact", whereas there is a lot of variation and uncertainty around relatedness estimates. I think this could be alleviated by just including a sentence in the caption saying something like "This diagram was constructed to be consistent with the analyses conducted; however, there is uncertainty around relatedness estimates and thus around some of these relationships".

14. Line 645-646: The authors say that Caroline "rarely took care of" Alexander, even though she was his mom. This seems to strong. Do the authors mean that she was "rarely associated" with him? Those could be two very different things, and my guess is that the authors cannot say anything about how good of a mother she is. Similarly in line 650: I would suggest removing "taking care of" and replace it with what the actually observed (e.g., "associated with").

15. I think the map (Figure S1) should be in the paper rather than in the supplementary material.

Review form: Reviewer 2

Is the manuscript scientifically sound in its present form?

Yes

Are the interpretations and conclusions justified by the results?

Yes

Is the language acceptable?

Yes

Do you have any ethical concerns with this paper?

No

Have you any concerns about statistical analyses in this paper?

No

Recommendation?

Accept with minor revision (please list in comments)

Comments to the Author(s)

Sarano et al. have done an excellent job responding to comments from both reviewers. I have a few additional minor comments below, but otherwise look forward to seeing this manuscript in press.

Line 81: add scientific names for long and short-finned pilot whales

Line 200: Please add here how many individuals have been photo identified in the population, whether there is an abundance estimate for the local population, and if so what it is.

Line 421: should this be " \leq " ?

Decision letter (RSOS-201794.R0)

Dear Dr Jung

The Editors assigned to your paper RSOS-201794 "Kin relationships in cultural species of the marine realm: case study of a matrilineal social group of sperm whales off Mauritius Island, Indian Ocean" have now received comments from reviewers and would like you to revise the paper in accordance with the reviewer comments and any comments from the Editors. Please note this decision does not guarantee eventual acceptance.

Please submit your revised manuscript and required files (see below) no later than 21 days from today's (ie 25-Nov-2020) date. Note: the ScholarOne system will 'lock' if submission of the revision is attempted 21 or more days after the deadline. If you do not think you will be able to meet this deadline please contact the editorial office immediately.

Please note article processing charges apply to papers accepted for publication in Royal Society Open Science (<https://royalsocietypublishing.org/rsos/charges>). Charges will also apply to papers transferred to the journal from other Royal Society Publishing journals, as well as papers submitted as part of our collaboration with the Royal Society of Chemistry

(<https://royalsocietypublishing.org/rsos/chemistry>). Fee waivers are available but must be requested when you submit your revision (<https://royalsocietypublishing.org/rsos/waivers>).

on behalf of Prof Pete Smith (Subject Editor)
openscience@royalsociety.org

Associate Editor Comments to Author:

The reviewers are largely happy with the scientific/analytical changes you have implemented, but there are a number of linguistic/typographic changes recommended that would improve the work. We'd like you to take advantage of this opportunity to tweak the language and punctuation to improve the readability - you might benefit from the use of a language editing service such as those at <https://royalsociety.org/journals/authors/benefits/language-editing/>. We'll look forward to reading your final version soon.

Reviewer comments to Author:

Reviewer: 1

Comments to the Author(s)

The manuscript is improved from the first draft; however, there is still quite a bit of room for improvement, although perhaps it is more cosmetic this time. I found in some parts, particularly the Abstract, but also elsewhere, the work was over-sold and the wording should be toned down. Additionally, I think the context of our knowledge of sperm whale social structure is not represented accurately in the Introduction, which provides a false context for this paper. Other than those, I have several more specific suggestions to improve the paper, as described below.

1. Line 48-49: I think it is overselling it to call the sampling technique "innovative" (it is just collecting skin that is seen floating in the water after the animals pass, after all). I suggest changing this to "using underwater observations, individual-specific identification, and genetic analyses based on mitochondrial sequencing and microsatellite profiling."
2. Lines 53-55: This statement is incorrect. Konrad et al. (2018) did the same thing for multiple groups in their paper (some of which were completely sampled).
3. Line 57: I think the word "adoption" here could be misleading. Technically, this would mean that a mother nurses and raises a calf that is not her own (which has been reported in whales). But it is not clear that that happened here, and it would be easy for readers to get confused. I would use clearer wording, perhaps "We highlight a likely case of an unrelated female being integrated into a social unit, presenting a mtDNA..."
4. Lines 81-83: I think "strongly dependent on kin relationships" is an over-statement. For some of these species it is definitely true, but I think the jury is still out on the importance of relatedness for some of them (e.g., long-finned pilot whales).

5. Line 91: It would be helpful if the authors define what is meant by a “fission-fusion model” so that all of the readers are on the same page.

6. Line 104: This isn't quite right. These aren't “genetically distinct” populations. It is true that maternally-direct site-fidelity in baleen whales leads to distinct structure of mitochondrial haplotypes between whales in different feeding grounds. But often whales from different feeding grounds all breed together in the winter. Thus, the feeding groups represent seasonal population structure, but certainly not “genetically distinct populations”.

7. Lines 171-172: The statement that “all samples used in these studies were biopsies...” is not true. Many are based on sloughed skin.

8. Lines 173-177: This doesn't provide an accurate picture of the studies underlying our current knowledge of sperm whale social structure. It is very hand-wavy and inaccurate. This may be true for some populations, but not for others. Painting these studies in this way sets them up as a straw-man, presumably to make this current study look better. This does not seem appropriate.

9. Lines 222-224: I think it is over-selling it. Although perhaps true for this particular geographical region, social structure in sperm whales has been well-studied other places. This sentence also does not add anything to the paper.

10. Lines 292-306: It is a bit weird that some description of the mitochondrial sequencing and microsatellite genotyping are provided here, but then other aspects of these procedures are described separately, under their own subheadings. It would be clearer if the text from here was moved to its appropriate subheading, so that all of the information describing each procedure was kept together. Also on line 292 it states that 774 bp of the control region was amplified, but on line 310 it says 638. Why the difference?

11. Line 356: More information should be provided about how this error rate of 2.2% was calculated. Is it per allele, per locus, per genotype? Exactly how many discrepancies were identified? The authors should be very explicit in this explanation. I had this comment last time, and the authors address it in their response, but it should be added to the manuscript.

12. Lines 512-523: For adult females that were already adult when the study started, and that showed parent-offspring relatedness with each other, how did they authors decide who was the mother and who was the offspring (since all were adult upon the initiation of the study). This should be clearly explained.

13. Figure 2: I think the diagram is an intuitive and informative way to visualize the putative relationships. However, my concern is that they could be interpreted as “fact”, whereas there is a lot of variation and uncertainty around relatedness estimates. I think this could be alleviated by just including a sentence in the caption saying something like “This diagram was constructed to be consistent with the analyses conducted; however, there is uncertainty around relatedness estimates and thus around some of these relationships”.

14. Line 645-646: The authors say that Caroline “rarely took care of” Alexander, even though she was his mom. This seems too strong. Do the authors mean that she was “rarely associated” with him? Those could be two very different things, and my guess is that the authors cannot say anything about how good of a mother she is. Similarly in line 650: I would suggest removing “taking care of” and replace it with what was actually observed (e.g., “associated with”).

15. I think the map (Figure S1) should be in the paper rather than in the supplementary material.

Reviewer: 2

Comments to the Author(s)

Sarano et al. have done an excellent job responding to comments from both reviewers. I have a few additional minor comments below, but otherwise look forward to seeing this manuscript in press.

Line 81: add scientific names for long and short-finned pilot whales

Line 200: Please add here how many individuals have been photo identified in the population, whether there is an abundance estimate for the local population, and if so what it is.

Line 421: should this be "<=" ?

===PREPARING YOUR MANUSCRIPT===

===PREPARING YOUR REVISION IN SCHOLARONE===

Author's Response to Decision Letter for (RSOS-201794.R0)

See Appendix C.

RSOS-201794.R1 (Revision)

Review form: Reviewer 1

Is the manuscript scientifically sound in its present form?

Yes

Are the interpretations and conclusions justified by the results?

Yes

Is the language acceptable?

Yes

Do you have any ethical concerns with this paper?

No

Have you any concerns about statistical analyses in this paper?

No

Recommendation?

Accept with minor revision (please list in comments)

Comments to the Author(s)

The authors have done a good job revising their manuscript and responding to reviewer comments. One main comment I have remaining is data accessibility. The authors only make their mitochondrial sequences available, but the microsatellite data are not. The majority of the paper hinges on the microsatellite data, and therefore those need to be available if the work is to be replicable. I also have two minor comments below.

1. First, I still don't agree with the statement that maternally directed site-fidelity in humpback whales results in "genetically-distinct populations at very small geographical scales." (lines 109-111). It certainly leads to populations structuring at small scales (indicated by the papers cited, among others), but these are not genetically distinct populations. There is wide gene flow among humpback whales that utilize difference feeding/breeding areas, and they cannot be classified as distinct populations, and no one does (see listings in all relevant countries). The authors should change to "This maternally-directed fidelity leads, for instance, to population structure at very small geographical scales..."

2. Lines 118-119: The statement that "Boat observations only allow occasional identification of individuals..." is misleading. Indeed, the vast majority of marine mammal research is based on this. Many species can be very predictably and reliably studied with this approach, and it is hardly that individuals are 'only occasionally identified'. The authors even cite "pioneering work" on cultural transmission that is based on photo-identification. The authors should re-frame this, perhaps saying that although photo-identification is useful, it does not provide relatedness information (except for mother-calf pairs). Therefore, genetic analyses must be paired with photo-identification efforts to truly understand the relationship between kinship and social structure in marine mammals.

Decision letter (RSOS-201794.R1)

Dear Dr Jung

The Editors assigned to your paper RSOS-201794.R1 "Kin relationships in cultural species of the marine realm: case study of a matrilineal social group of sperm whales off Mauritius Island, Indian Ocean" have now received comments from reviewers and would like you to revise the paper in accordance with the reviewer comments and any comments from the Editors. Please note this decision does not guarantee eventual acceptance.

Please submit your revised manuscript and required files (see below) no later than 21 days from today's (ie 06-Jan-2021) date. Note: the ScholarOne system will 'lock' if submission of the revision is attempted 21 or more days after the deadline. If you do not think you will be able to meet this deadline please contact the editorial office immediately.

on behalf of Prof Pete Smith (Subject Editor)
openscience@royalsociety.org

Associate Editor Comments to Author:

Thank you for your patience over the festive period. The reviewer who has commented observes that there are a couple of points outstanding that you need to address before we can accept the paper.

Particularly problematic is the absence of part of the dataset - this needs to be made accessible before we can accept the paper (if it is already accessible, please make sure this is clarified in the data access statement of the paper).

There remains a slight concern regarding the clarity of the writing in the paper - there may be an element of personal taste in the styling of parts of the language in the paper, but we would encourage you to take a final look at this in case any additional clarity can be introduced through careful editing.

Reviewer comments to Author:

Reviewer: 1

Comments to the Author(s)

The authors have done a good job revising their manuscript and responding to reviewer comments. One main comment I have remaining is data accessibility. The authors only make their mitochondrial sequences available, but the microsatellite data are not. The majority of the paper hinges on the microsatellite data, and therefore those need to be available if the work is to be replicable. I also have two minor comments below.

1. First, I still don't agree with the statement that maternally directed site-fidelity in humpback whales results in "genetically-distinct populations at very small geographical scales." (lines 109-111). It certainly leads to populations structuring at small scales (indicated by the papers cited, among others), but these are not genetically distinct populations. There is wide gene flow among humpback whales that utilize difference feeding/breeding areas, and they cannot be classified as distinct populations, and no one does (see listings in all relevant countries). The authors should change to "This maternally-directed fidelity leads, for instance, to population structure at very small geographical scales..."

2. Lines 118-119: The statement that "Boat observations only allow occasional identification of individuals..." is misleading. Indeed, the vast majority of marine mammal research is based on this. Many species can be very predictably and reliably studied with this approach, and it is hardly that individuals are 'only occasionally identified'. The authors even cite "pioneering work" on cultural transmission that is based on photo-identification. The authors should re-frame this, perhaps saying that although photo-identification is useful, it does not provide relatedness information (except for mother-calf pairs). Therefore, genetic analyses must be paired with photo-identification efforts to truly understand the relationship between kinship and social structure in marine mammals.

===PREPARING YOUR MANUSCRIPT===

Please ensure that you include an acknowledgements' section before your reference list/bibliography. This should acknowledge anyone who assisted with your work, but does not

qualify as an author per the guidelines at <https://royalsociety.org/journals/ethics-policies/openness/>.

===PREPARING YOUR REVISION IN SCHOLARONE===

- Ensure that your data access statement meets the requirements at <https://royalsociety.org/journals/authors/author-guidelines/#data>. You should ensure that you cite the dataset in your reference list. If you have deposited data etc in the Dryad repository, please include both the 'For publication' link and 'For review' link at this stage.
- If you are requesting an article processing charge waiver, you must select the relevant waiver option (if requesting a discretionary waiver, the form should have been uploaded at Step 3 'File upload' above).
- If you have uploaded ESM files, please ensure you follow the guidance at <https://royalsociety.org/journals/authors/author-guidelines/#supplementary-material> to include a suitable title and informative caption. An example of appropriate titling and captioning may be found at https://figshare.com/articles/Table_S2_from_Is_there_a_trade-off_between_peak_performance_and_performance_breadth_across_temperatures_for_aerobic_sc_ope_in_teleost_fishes_/3843624.

Author's Response to Decision Letter for (RSOS-201794.R1)

See Appendix D.

Decision letter (RSOS-201794.R2)

Dear Dr Jung,

It is a pleasure to accept your manuscript entitled "Kin relationships in cultural species of the marine realm: case study of a matrilineal social group of sperm whales off Mauritius Island, Indian Ocean" in its current form for publication in Royal Society Open Science.

on behalf of Prof Pete Smith (Subject Editor)
openscience@royalsociety.org

Appendix A

Abstract

52-53 – remove “thus confirming matrilineality of the group” (see below)

57 – remove Asian elephants (see below)

Introduction

93-94: The introduction should include at least a brief acknowledgement of what studies have been conducted, and what is known, about kinship, social structure, and population structure in cetacean species. Although social behaviors are touched upon in lines 74-82, there is little/no discussion of existing studies that examine links between kinship and social behavior in cetaceans, including species such as the false killer whale, long- and short-finned pilot whales, right whale, and beluga whale. An incomplete list of some examples that should be included:

Carroll, E. L., Baker, C. S., Watson, M., Alderman, R., Bannister, J., Gaggiotti, O. E., Gröcke, D. R., Patenaude, N., & Harcourt, R. (2015). Cultural traditions across a migratory network shape the genetic structure of southern right whales around Australia and New Zealand. *Scientific Reports*, 5(1), 16182. <https://doi.org/10.1038/srep16182>

Martien, K., Taylor, B., Chivers, S., Mahaffy, S., Gorgone, A., & Baird, R. (2019). Fidelity to natal social groups and mating within and between social groups in an endangered false killer whale population. *Endangered Species Research*, 40, 219–230. <https://doi.org/10.3354/esr00995>

Martien, K. K., Chivers, S. J., Baird, R. W., Archer, F. I., Gorgone, A. M., Hancock-Hanser, B. L., Mattila, D., McSweeney, D. J., Oleson, E. M., Palmer, C., Pease, V. L., Robertson, K. M., Schorr, G. S., Schultz, M. B., Webster, D. L., & Taylor, B. L. (2014). Nuclear and Mitochondrial Patterns of Population Structure in North Pacific False Killer Whales (*Pseudorca crassidens*). *Journal of Heredity*, 105, 611–626. <https://doi.org/10.5061/dryad.2pq32>

Baird, R., Hanson, M., Schorr, G., Webster, D., McSweeney, D., Gorgone, A., Mahaffy, S., Holzer, D., Oleson, E., & Andrews, R. (2012). Range and primary habitats of Hawaiian insular false killer whales: informing determination of critical habitat. *Endangered Species Research*, 18(1), 47–61. <https://doi.org/10.3354/esr00435>

Mahaffy, S. D., Baird, R. W., McSweeney, D. J., Webster, D. L., & Schorr, G. S. (2015). High site fidelity, strong associations, and long-term bonds: Short-finned pilot whales off the island of Hawai'i. *Marine Mammal Science*, 31(4), 1427–1451. <https://doi.org/10.1111/mms.12234>

Van Cise, A. M., Martien, K. K., Mahaffy, S. D., Baird, R. W., Webster, D. L., Fowler, J. H., Oleson, E. M., & Morin, P. A. (2017). Familial social structure and socially driven genetic differentiation in Hawaiian short-finned pilot whales. *Molecular Ecology*, 26(23), 6730–6741. <https://doi.org/10.1111/mec.14397>

Van Cise, A. M., Mahaffy, S. D., Baird, R. W., Mooney, T. A., & Barlow, J. (2018). Song of my people: dialect differences among sympatric social groups of short-finned pilot whales in Hawai'i. *Behavioral Ecology and Sociobiology*, 72(12), 193.
<https://doi.org/10.1007/s00265-018-2596-1>

Rendell, L., Cantor, M., Gero, S., Whitehead, H., & Mann, J. (2019). Causes and consequences of female centrality in cetacean societies. In *Philosophical Transactions of the Royal Society B: Biological Sciences* (Vol. 374, Issue 1780).
<https://doi.org/10.1098/rstb.2018.0066>

Corry-crowe, G. O., Suydam, R., Quakenbush, L., Potgieter, B., Harwood, L., Litovka, D., Ferrer, T., Citta, J., Burkanov, V., Frost, K., & Mahoney, B. (2018). *Migratory culture , population structure and stock identity in North Pacific beluga whales (Delphinapterus leucas)*. <https://doi.org/10.5061/dryad.6b70g11.Microsatellite>

Palsbøll, P. J., Heide-Jørgensen, M. P., & Bérubé, M. (2002). Analysis of mitochondrial control region nucleotide sequences from Baffin Bay beluga, (*Delphinapterus leucas*): detecting pods or sub-populations? *NAMMCO Scientific Publications*, 4, 39.
<https://doi.org/10.7557/3.2836>

Oremus, M., Gales, R., Kettles, H., & Baker, C. S. (2013). Genetic Evidence of Multiple Matrilines and Spatial Disruption of Kinship Bonds in Mass Strandings of Long-finned Pilot Whales, *Globicephala melas*. *Journal of Heredity*, 104(3), 301–311.
<https://doi.org/10.1093/jhered/est007>

99-100: incomplete sentence

101-103: This sentence is vague – please explain what “first importance” means, and describe the pioneering works referred to in the sentence and how they demonstrate that cultural behavior is important.

105: “mammal” should be “mammalian”

116: “valuable” – do you mean “plausible”?

120: Please explain the connection between sperm whale body size/anatomy and cultural traditions, or remove the seemingly superfluous mention that they are large animals.

148: Here, instead of samples “taken from the sea surface”, it would be good to state directly that you are referring to tissue biopsies, most often collected using a crossbow dart. In most genetic studies, the biopsied animal is easily identified through photographs – if that isn't the case here, it will be helpful for you to explain why.

154-163: Much of this section is superfluous and can be shortened. Most of it belongs in the methods, rather than the introduction.

154: It would be good to have more information about the local sperm whale population – how big is it? What geographic area does it cover? What portion of their range does your study area comprise? Is Irene’s group the only resident social group? If so, what portion of the year are other individuals or social groups present? Any background information you have will provide helpful context about the social group and this study.

159: “underwater observations” – using AUVs? Pole-mounted Go-Pros? Please describe the underwater observation systems used in this study in detail in the methods section.

163-172: If these results haven’t been published anywhere previously, they belong in the results section, and the associated photo ID methods need to be included in the methods section. If these results have been previously published, they need to be properly cited.

163: which morphological characteristics? More detail needed here, and citations.

173-175: This also belongs in the methods sections. Also, please explain how you assessed confidence in which skin fragments belong to which individual.

Methods

199-201: What platform was used for the field work? Small boat? Tour ships? Research Vessel?

203-205: How were the underwater observations collected? Diver? AUV? Pole-mounted recorders? How close were the recorders to the animals at the time of recording?

206-214: It is important to the study to detail what external morphological characteristics were used to identify individuals.

215-222: This is not methods, it belongs in the results section.

226: Please explain why sloughed skin was used rather than more traditional tissue biopsies.

240: How close did snorkelers get to the whales?

268: “amplified and analyzed as described” – this seems out of place?

302: Please define Pid and describe how it is calculated.

316-317: Please describe the method used to calculate error rate.

326: What was done after the first investigation? Were mis-matched samples removed from the study? Or was an attempt made to match those samples to the “correct” individual? If so, what was the protocol for re-matching DNA sample to a different individual?

332: Do you specifically mean avuncular here,

Results

405-408: Please include details about any discrepancies among individuals that may have been due to genotyping error or other sample processing error. I.e., did genotypes of all genetic individuals match

100%? What was the rate of mismatch among samples within an individual? If any mismatch was identified, was it found in the mtDNA haplotype or in the microsatellites?

422-425: This should be in the methods section.

452: This number seems very low, given the high levels of relatedness among most individuals within the group. Average pairwise relatedness is higher in other studies of highly-social mammals, e.g. among female elephants within a core group relatedness was 0.15 (Archie et al. 2006); average pairwise relatedness within pilot whale social units was ~0.18 (Van Cise et al. 2017). The review by Briga et al. (2012) cited in this study reports average relatedness among female sperm whales within a group to be 0.19 (reported from Quellar and Goodnight (1989)). How was average relatedness within groups calculated? It would be good to report relatedness for all dyads in the study in a supplemental table. The discussion should include some explanation for the low within-group relatedness values reported here, compared to previously reported values for sperm whales and other social cetaceans.

458: What about Vanessa? Figure 2 shows no first or second degree relationships between Vanessa and other members of the group, yet she is one of the oldest females in the group. Does Vanessa have the same haplotype as the rest of the group members (SW_M), despite an apparent lack of kinship with the group? Further discussion of Vanessa, both in the results and the discussion sections, seems warranted.

469: “observed only 28 times” – please specify what years those sightings occurred in.

491-496: Supporting information needs to accompany this section, if you are going to include it, e.g. table of dyad relatedness calculations for these two individuals with the others in the study. It would also be good to include these individuals in Figure 2, specifying that they are from a different social group.

Discussion

533-536: This is too strong a statement based on the results of the current study. This study examines a single social group found off the coast of Mauritius – its scope is limited. The authors should base this statement on a larger sample size covering a greater portion of the local population. If there are published data on mtDNA CR haplotype frequencies in the larger population, those can be used to provide context to the results for this specific social group. However, it cannot be assumed *a priori* that the prevalent mtDNA CR haplotype in this study isn't common in the local population just because it isn't common in other parts of the Indian Ocean.

570: Asian elephants have an $r = 0.37$ (Briga et al. 2017 reporting results from Quellar and Goodnight), so should be removed from this list.

574-579: Please define both social relational complexity and organisational complexity, and discuss any implications of this framework on our understanding of sperm whale socio-genetic structure.

580-599: These are very interesting observations – some attempt to quantify this behavior should be undertaken, either as part of this study or a future study. It would be useful to compare dyad relatedness and probable kinship relationships with an overall index of pairwise association among individuals or with an index of the rate of nursing or other care-type behaviors, in order to determine the importance of alloparental care within the group.

595-599: This statement is unclear, in part because organisational complexity hasn't yet been defined. Please describe why alloparental care is more consistent with organizational complexity. Please also clarify the link you are trying to make, between the positive phylogenetic signal of alloparental care in mammals, and sperm whales having a relational complexity.

663-664: Again, this statement is strong given the sample size, and lack of sample coverage from other social groups in the population. Microsatellite markers indicated low within-group relatedness compared to other studies. Several of the adult females (e.g. Vanessa, Claire) have no relationship to other animals in the group. These findings do not support "strict matrilineality" as stated here.

697: Replace "most" with "more" – many marine species exhibit socially transmitted cultural behaviors and socially driven genetic structure.

Figure 2: The figure caption indicates that mitochondrial haplotypes are displayed, but they are not shown in the figure.

Appendix B

Answers to reviewer comments.

Our answers are in blue.

Reviewers' Comments to Author:

Reviewer: 1

Larger-Scale Comments :

1. As a whole, the entire section describing different aspects of social structure, and the relative strengths and weaknesses of group living could use stronger theoretical foundations. Some of these are listed below, but this whole section could be strengthened.

The whole paragraph has been removed (L83-L92), as we also removed our results and discussion about the average relatedness in the Irène's group (following Rq. 5 of reviewer 1). More examples of what is known about cetacean kinship and social structure has been added (second paragraph of the introduction)

- a. Lines 86-88: This idea needs explanation. The authors should clarify what they mean by, and explain the underlying theory, behind the statement "Low average relatedness...could be correlated in particular to high individual social relationships disparities.." I don't follow this at all.

Sentence removed

- b. Lines 91-92: The statement "Group size and relatedness are also correlated, in a species-specific manner" needs much more explanation/justification.

Sentence removed

2. The context of what is known about sperm whale social groups does not seem to be properly represented, and therefore this whole section could use revision. For example Lines 140-153 give the impression that this is not well known, and that, for some unknown reason, different studies produce different results. However, this is not necessarily true, and is certainly inaccurate. Instead, we know that the social structure of sperm whale female units differs fairly substantially in different regions of the world (e.g., Whitehead & Kahn (1992) Can. J. Zool. 70: 2145-2149, Whitehead (2003) Sperm Whales: Social Evolution in the Ocean). Therefore, we would expect to obtain different results due to the differences in underlying biology in different areas, rather than due to some sort of analytical discrepancy.

We acknowledge this point and a new paragraph has been added there that describes some main geographical differences between Atlantic and Pacific sperm whale social groups. The last sentences of the paragraph have also been rephrased to lower their significance

3. The importance of underwater observations. The authors suggest that this aspect of their work is quite important, but I don't ever see this justified satisfactorily. Why, or under what conditions, are underwater observations better? The logistics seem difficult to me. If a group of 28 sperm whales swims by, on how many occasions can a snorkeler get identification photographs of all 28? Collecting data from the water seems fine, but is not adequately justified here. The authors should thoroughly explain why they are taking this approach rather than vessels.

We have described the results of our underwater observation works in another manuscript, which is presently submitted, and that we included in this submission as "response to referees", and which is cited in the present manuscript (ref 54). "Underwater observation allows for instance to distinguish juveniles, who rarely fluke and whose caudal fin often present no distinctive mark. More generally, underwater observation strongly increases the number of distinctive marks per individual."

4. Not enough information is provided on the genetic mismatches/genotyping errors.

a. Lines 300-302: How many alleles could differ between two genotypes and have them still be considered "similar". This should be explicit, and well-justified.

Samples from a same individual were identified using the Identity Analysis in CERVUS, which determines the probability (pID) of a same genotype to occur in two unrelated individuals, given allele frequencies. The analysis was conducted allowing up to 2 divergent loci to consider possible genotyping errors (i.e. null, missing or discordant alleles), though probabilities are only generated for pairs which match at all compared loci. Samples with more than 2 divergent loci (n=2) were not considered. Sample pairs presenting one or two divergent loci were further analyzed: In all the cases, a null allele was involved (one locus was apparently homozygous for a sample, and heterozygous for the other sample). In any case divergent alleles were observed.

b. Lines 302-304: Once "genetic individuals" were identified, which genotype was used as representative of that individual, and how was that decided? This could have big implications for the relatedness analyses.

Fuzzy matching genotypes were only due to the presence of a null allele at a locus, no discordant alleles were notified. Exact matching genotypes and fuzzy matching ones were assumed to correspond to skin samples taken at different occasions from a same individual and define a *genetic individual* (see Table S1). When one genotype of an individual had an allele clearly detected in some PCRs and not in another PCR (a null allele), that allele was retained in the final genetic profile of the individual. No other corrections were done to determine the genotypes of the "genetic individuals".

c. What are the units for this error rate. Is this per amplification, per allele, per locus, or per genotype? These all mean very different things, but it is not clear what is being referred to here, or how this number was obtained.

Seventeen individuals were genotyped on at least 3 independent DNA extracts (Table S1). These different samples of a same individual allowed us to perform replicate PCRs for all the nuclear loci and to estimate the genotyping errors linked to possible poor-quality DNA extracts. All loci (n=14) have 61 PCR replicas for which 38 out of 1708 alleles were incorrect (or 25 out of 854 loci were incorrect). We calculated an overall per-allele error rate of 2.2% which was subsequently used in kinship analysis.

5. Interpretation of relatedness analyses. All relatedness estimates are based on comparison to a “reference group”, which is usually all of the genotypes of the study (unless otherwise specified). All relatedness estimators are standardized so that the average relatedness within a sample set is 0 (or approximately so). Because this group consisted of genotypes from one social unit, it is not appropriate for estimating the average relatedness within that unit (it will be about zero). To do this, many more genotypes, from other social groups, would be needed. Then it would be possible to say something about the relatedness of individuals within this group relative to the population as a whole. However, since all but two genotypes are from individuals within this group, it is not appropriate to use these data to estimate the average relatedness within the group. Therefore, the estimate obtained (average $r = 0.048$) does not mean anything, because the appropriate reference group has not been sampled. Therefore, all aspects of this should be removed from the Results and Discussion. It is still appropriate to use these data to identify specific relationships among pairs of individuals, but even here their estimated r -values will be lower than what they truly are, again because there is not an appropriate reference group for comparison.

We agree with the reviewer, and we thank him for this analysis. We removed all discussions about the average relatedness, excepted to compare its value between the Irene’s group and the two subgroups.

Smaller-scale edits

1. Line 49: The statement that “Genotypes matched field identifications” is not accurate. There were some cases where this was not true, at least initially, and also such a statement is not possible (a genotype can’t match a field identification). I would just remove this sentence.

The sentence has been removed

2. Line 75: Should be *Physeter macrocephalus* (!)

Corrected. Sorry for this mistake

3. Lines 96-97: The statement “...most individuals are undistinguishable” requires more justification. I know of many cetacean species for which every individual can be identified, and therefore this statement does not seem accurate.

The sentence has been modified

4. Lines 174-175: “..attributed with enough confidence...” what does this mean? This statement requires much more explanation.

This part of the sentence has been removed

5. Line 99-100: I don't see how the previous sentences, regarding photo-ID and genetic sampling, lead to interest in social groups. This tie should be more clear.

This paragraph is an attempt to explain that, although there are specific difficulties to study marine mammals at sea, their social organisation draws a lot of attention from the scientific community. The previous sentence has been modified to reinforce this idea.

6. Line 103: What is meant by “of first importance”?

The sentence has been rephrased to clarify its meaning

7. Line 105: “nowadays” seems too informal for a scientific publication.

The sentence has been rephrased

8. Line 120: What is meant by “outsize body”? Perhaps different wording.

The sentence has been rephrased

9. Line 133-135: This sentence should be removed. The fact that no migrations were known, until recently, does not add to the paper.

Sentence removed

10. Line 248: CITES requires both an import and an export permit, yet this is just one number. What is the other one?

The export permit number has been added in the text

11. Line 251: I thought that biopsy samples were not collected (that all were sloughed skin).

“Biopsies” has been removed

12. Line 258: What concentration was the DNA samples standardized to?
10ng/μL (added in the text)

13. Lines 265-272: More details regarding microsatellite amplification are required: annealing temperatures, were they amplified singly or in multiplex reactions, primer concentrations, fluorescent labels, etc. This could be a table in the Supplementary Material. Moreover, how were they size-separated and visualized? What sort of equipment?

The paragraph has been completed, and a supplementary table S2 presenting the amplification conditions has been added

14. Lines 546-549: This statement is not true. Konrad et al. (ref. 60) used genetic data to show that some individuals in social groups did not have close genetic ties therein, much like “Claire” here.

We modified the sentence.

In our understanding, Konrad et al. (2018) showed in fact that some individuals presented no genetically detectable level of kin relationships with other members of a same social group, but they did share the same mitochondrial haplotype. We discuss about the possibility that some individuals of a same matrilineal group do not present high (detectable) genetic relationships (third paragraph of “The Irène’s sperm whale group is matrilineal and shows extensive allomaternal care” in the discussion). Therefore, in our understanding, there is no clear evidence that the individuals described in Konrad et al. (2018) are mergers from other social groups.

Here, Claire has no close kin in the group, but moreover, she is the only one in the Irène’s group to possess a different mitochondrial haplotype. This point is strongly supporting the transfer from another group.

15. Line 568: Should be Clutton-Brock.

Reference removed (average relatedness is no more discussed)

16. Lines 568-579: This does not seem relevant to this study.

The whole paragraph has been removed following the recommendation of the reviewer

17. Lines 603-604: This sentence requires justification. Why is understanding this of major importance? To who?

The sentence has been clarified

18. Line 663-666: Should cite ref. 60 again here, as Konrad et al. characterized the matrilineal nature of sperm whale social groups in a different ocean basin.

Reference added.

19. Lines 697-705: Not sure this is immediately relevant to this study.

This last paragraph explicit main points of high interest we, and certainly others, would like to address in our future works. We would like to keep it as it is.

Reviewer: 2

Abstract

52-53 – remove “thus confirming matrilineality of the group” (see below)

We added “almost certainly” to lower the significance of the sentence. Social groups can be considered matrilineal when most females (or all offspring) remain, for life, with their mothers and other close female relatives. Taken together, the observation of the same individuals over a long time period, the mtDNA haplotype homogeneity and the number of first- and second-degree relationships are in agreement with this definition and have highlighted the matrilineality of this social group

57 – remove Asian elephants (see below)

Following the remark 5 of reviewer 1, and because of the absence of a reference group in our study, we removed from our manuscript all the analysis of the average r value of the Irene’s group.

Introduction

93-94: The introduction should include at least a brief acknowledgement of what studies have been conducted, and what is known, about kinship, social structure, and population structure in cetacean species. Although social behaviors are touched upon in lines 74-82, there is little/no discussion of existing studies that examine links between kinship and social behavior in cetaceans, including species such as the false killer whale, long- and short-finned pilot whales, right whale, and beluga whale. An incomplete list of some examples that should be included:

Carroll, E. L., Baker, C. S., Watson, M., Alderman, R., Bannister, J., Gaggiotti, O. E., Gröcke, D. R., Patenaude, N., & Harcourt, R. (2015). Cultural traditions across a migratory network shape the genetic structure of southern right whales around Australia and New Zealand. *Scientific Reports*, 5(1), 16182. <https://doi.org/10.1038/srep16182>

Martien, K., Taylor, B., Chivers, S., Mahaffy, S., Gorgone, A., & Baird, R. (2019). Fidelity to natal social groups and mating within and between social groups in an endangered false killer whale population. *Endangered Species Research*, 40, 219–230. <https://doi.org/10.3354/esr00995>

Martien, K. K., Chivers, S. J., Baird, R. W., Archer, F. I., Gorgone, A. M., Hancock-Hanser, B. L., Mattila, D., McSweeney, D. J., Oleson, E. M., Palmer, C., Pease, V. L., Robertson, K. M., Schorr, G. S., Schultz, M. B., Webster, D. L., & Taylor, B. L. (2014). Nuclear and Mitochondrial Patterns of Population Structure in North Pacific False Killer Whales (*Pseudorca crassidens*). *Journal of Heredity*, 105, 611–626. <https://doi.org/10.5061/dryad.2pq32>

Baird, R., Hanson, M., Schorr, G., Webster, D., McSweeney, D., Gorgone, A., Mahaffy, S., Holzer, D., Oleson, E., & Andrews, R. (2012). Range and primary habitats of Hawaiian insular false killer whales: informing determination of critical habitat. *Endangered Species Research*, 18(1), 47–61. <https://doi.org/10.3354/esr00435>

Mahaffy, S. D., Baird, R. W., McSweeney, D. J., Webster, D. L., & Schorr, G. S. (2015). High site fidelity, strong associations, and long-term bonds: Short-finned pilot whales off the island of Hawai‘i. *Marine Mammal Science*, 31(4), 1427–1451. <https://doi.org/10.1111/mms.12234>

Van Cise, A. M., Martien, K. K., Mahaffy, S. D., Baird, R. W., Webster, D. L., Fowler, J. H., Oleson, E. M., & Morin, P. A. (2017). Familial social structure and socially driven genetic differentiation in Hawaiian short-finned pilot whales. *Molecular Ecology*, 26(23), 6730– 6741. <https://doi.org/10.1111/mec.14397>

Van Cise, A. M., Mahaffy, S. D., Baird, R. W., Mooney, T. A., & Barlow, J. (2018). Song of my people: dialect differences among sympatric social groups of short-finned pilot whales in Hawai'i. *Behavioral Ecology and Sociobiology*, 72(12), 193. <https://doi.org/10.1007/s00265-018-2596-1>

Rendell, L., Cantor, M., Gero, S., Whitehead, H., & Mann, J. (2019). Causes and consequences of female centrality in cetacean societies. In *Philosophical Transactions of the Royal Society B: Biological Sciences* (Vol. 374, Issue 1780). <https://doi.org/10.1098/rstb.2018.0066>

Corry-crowe, G. O., Suydam, R., Quakenbush, L., Potgieter, B., Harwood, L., Litovka, D., Ferrer, T., Citta, J., Burkanov, V., Frost, K., & Mahoney, B. (2018). Migratory culture , population structure and stock identity in North Pacific beluga whales (*Delphinapterus leucas*). <https://doi.org/10.5061/dryad.6b70g11.Microsatellite>

Palsbøll, P. J., Heide-Jørgensen, M. P., & Bérubé, M. (2002). Analysis of mitochondrial control region nucleotide sequences from Baffin Bay beluga, (*Delphinapterus leucas*): detecting pods or sub-populations? *NAMMCO Scientific Publications*, 4, 39. <https://doi.org/10.7557/3.2836>

Oremus, M., Gales, R., Kettles, H., & Baker, C. S. (2013). Genetic Evidence of Multiple Matrilines and Spatial Disruption of Kinship Bonds in Mass Strandings of Long-finned Pilot Whales, *Globicephala melas*. *Journal of Heredity*, 104(3), 301–311. <https://doi.org/10.1093/jhered/est007>

More examples of what is known about cetacean kinship and social structure has been added (second paragraph of the introduction). We thank the reviewer for this relevant list of publications.

99-100: incomplete sentence

The sentence has been completed to clarify its meaning

101-103: This sentence is vague – please explain what “first importance” means, and describe the pioneering works referred to in the sentence and how they demonstrate that cultural behavior is important.

The sentence has been modified (also following a remark from reviewer 1)

105: “mammal” should be “mammalian”

This has been corrected

116: “valuable” – do you mean “plausible”?

Yes, word modified

120: Please explain the connection between sperm whale body size/anatomy and cultural traditions, or remove the seemingly superfluous mention that they are large animals.

The sentence has been modified (following also a remark from reviewer 1)

148: Here, instead of samples “taken from the sea surface”, it would be good to state directly that you are referring to tissue biopsies, most often collected using a crossbow dart. In most genetic studies, the biopsied animal is easily identified through photographs – if that isn’t the case here, it will be helpful for you to explain why.

The sentence has been modified and the word “biopsies” is now used. The reason why sperm whales can be difficult to identify from a boat are developed in the manuscript submitted (Sarano et al.) which is joined to this submission.

154-163: Much of this section is superfluous and can be shortened. Most of it belongs in the methods, rather than the introduction.

We agree in part to this remark. But in this paragraph, we wanted to highlight the work done in the field by Mauritian and French NGOs. We would like to keep it for this reason

154: It would be good to have more information about the local sperm whale population – how big is it? What geographic area does it cover? What portion of their range does your study area comprise? Is Irene’s group the only resident social group? If so, what portion of the year are other individuals or social groups present? Any background information you have will provide helpful context about the social group and this study.

Done. A new paragraph has been added.

Methods:

All the field work protocols are now described in the other manuscript joined to this submission (Sarano et al. submitted). Some more information are given below

199-201: What platform was used for the field work? Small boat? Tour ships? Research Vessel?

The boat was a 15m cabin cruiser designed for divers and that has a regular mooring at Trou aux Biches (Mauritius)

4.1 Particulars of vessel:	
Name:	BLUE WATER DIVERS 4
Type/Class:	CABIN CRUISER
Nationality (Flag State):	MAURITIUS
Identification Number (IMO/Lloyds No.):	PC0724 OL 25
Website for diagram & specifications	www.bluewaterdivingcentre.com
Owner:	HUGUES VITRY AND ASSOCIATE
Operator:	BLUE WATER DIVING CENTRE LIMITED
Overall length (meters):	14,77 meters x 4,65 meters.
Maximum draught (meters):	1.20 meter
Displacement/Gross tonnage:	
Propulsion:	2 x 210 HP caterpillar turbo diesel
Cruising & maximum speed:	9 knot – 12 knot
Call sign:	
INMARSAT number and method and capability of communication (including emergency frequencies):	Garmin VHF Radio – Normal channel: 09 – Emergency channel:16
Name of master:	Axel Preud'homme / Hugues Vitry (responsible)
Number of crew:	2

203-205: How were the underwater observations collected? Diver? AUV? Pole-mounted recorders? How close were the recorders to the animals at the time of recording?

All underwater observations were video-recorded, either with a Sony F55 4K, a Sony EXIR HD, a Nikon D800 Camera in Hugyfot housing or a GoPro camera Hero 4, 7 and 8, by a scuba diver and snorkelers (see Sarano et al. submitted for a detailed description)

206-214: It is important to the study to detail what external morphological characteristics were used to identify individuals.

see Sarano et al. submitted for a detailed description of all the external marks used to identify the sperm whales

215-222: This is not methods, it belongs in the results section.

The paragraph is now at the beginning of the result section

226: Please explain why sloughed skin was used rather than more traditional tissue biopsies.

Sloughed skin sampling was used first because it allowed frequent individual-specific sampling (strongly more frequent than by taking biopsies with a crossbow for sperm whales). Moreover, sloughed skins are non-invasive samples.

240: How close did snorkelers get to the whales?

This is described in Sarano et al. submitted. Snorkelers were “passive” waiting for the sperm whales to approach. They can be some meter close to the whales when taking the skin samples

268: “amplified and analyzed as described” – this seems out of place?

The sentence has been rephrased

302: Please define P_{id} and describe how it is calculated.

Done

316-317: Please describe the method used to calculate error rate.

Described in detail in the answer to questions 4 of reviewer 1

326: What was done after the first investigation? Were mis-matched samples removed from the study?

Or was an attempt made to match those samples to the “correct” individual? If so, what was the protocol for re-matching DNA sample to a different individual?

The correlation between genetic individuals and field-identified individuals was done for each skin sample. In case of disagreement between the two identifications (a skin sample attributed to one individual in the field but presenting a genotype corresponding to another genetic individual, this occurred for 4 samples), the first investigation was to analyze more in detail the video recording of the taking of the skin sample, and to determine if the genetic individual could have been close to the sampling site, leading to a misidentification of the sloughed skin in the field. This was the case for the 4 misidentified samples, which were then reattributed to the correct individual and kept in the analyses.

332: Do you specifically mean avuncular here,

We do not clearly catch the point of this remark. We mean avuncular as “any of the four relationship categories involving uncles or aunts with nephews or nieces” (definition from Blouin, Trends in Ecology, 2003).

Results

405-408: Please include details about any discrepancies among individuals that may have been due to genotyping error or other sample processing error. I.e., did genotypes of all genetic individuals match

A new paragraph has been added, describing the differences observed between different samples of a same individual (null alleles in all the cases)

100%? What was the rate of mismatch among samples within an individual? If any mismatch was identified, was it found in the mtDNA haplotype or in the microsatellites?

No mismatch was found in the mitochondrial DNAs (all samples from a same individual were identical). For NuDNA, mismatches were only due to missing null alleles in the genotypes of some samples.

422-425: This should be in the methods section.

The paragraph has been moved to the methods section

452: This number seems very low, given the high levels of relatedness among most individuals within the group. Average pairwise relatedness is higher in other studies of highly-social mammals, e.g. among female elephants within a core group relatedness was 0.15 (Archie et al. 2006); average pairwise relatedness within pilot whale social units was ~0.18 (Van Cise et al. 2017). The review by Briga et al. (2012) cited in this study reports average relatedness among female sperm whales within a group to be 0.19 (reported from Quellar and Goodnight (1989)). How was average relatedness within groups calculated? It would be good to report relatedness for all dyads in the study in a supplemental table. The discussion should include some explanation for the low within-group relatedness values reported here, compared to previously reported values for sperm whales and other social cetaceans.

Following the remark 5 of reviewer 1, and because of the absence of a reference group in our study, we removed from our manuscript all the analysis of the average r value of the Irene's group.

We also added a supplementary table (supplementary table S4) listing relatedness values for all dyads.

458: What about Vanessa? Figure 2 shows no first or second degree relationships between Vanessa and other members of the group, yet she is one of the oldest females in the group. Does Vanessa have the same haplotype as the rest of the group members (SW_M), despite an apparent lack of kinship with the group? Further discussion of Vanessa, both in the results and the discussion sections, seems warranted.

Among all adult females, two, Claire and Vanessa, shared no first- or second-degree relationships with other members of the group, and another, Yukimi, shared only one second-degree relationship with an adult female. However, Vanessa and Yukimi have the same mitochondrial haplotype than the rest of the group. But Claire presented a different mitochondrial haplotype, thus reinforcing her marked genetic difference with the rest of the group.

A sentence about Yukimi and Vanessa has been added in the discussion

469: "observed only 28 times" – please specify what years those sightings occurred in.

In 2019 (added in the text)

491-496: Supporting information needs to accompany this section, if you are going to include it, e.g. table of dyad relatedness calculations for these two individuals with the others in the

study. It would also be good to include these individuals in Figure 2, specifying that they are from a different social group.

A new table (supplementary table S7) listing all the relatedness values between ClanReshna 1 and ClanReshna 2 and all members of the Irene's clan has been added

Discussion

533-536: This is too strong a statement based on the results of the current study. This study examines a single social group found off the coast of Mauritius – its scope is limited. The authors should base this statement on a larger sample size covering a greater portion of the local population. If there are published data on mtDNA CR haplotype frequencies in the larger population, those can be used to provide context to the results for this specific social group. However, it cannot be assumed a priori that the prevalent mtDNA CR haplotype in this study isn't common in the local population just because it isn't common in other parts of the Indian Ocean.

Sharing a common haplotype, minor in the Indian Ocean, is in line with the idea of a matrilineal social group. We have added a sentence about the requirement to obtain more data from the local population of sperm whales.

570: Asian elephants have an $r = 0.37$ (Briga et al. 2017 reporting results from Queller and Goodnight), so should be removed from this list.

574-579: Please define both social relational complexity and organisational complexity, and discuss any implications of this framework on our understanding of sperm whale socio-genetic structure.

Following the remark 5 of reviewer 1, and because of the absence of a reference group in our study, we removed from our manuscript all the analysis of the average r value of the Irene's group, including paragraph 568-579.

580-599: These are very interesting observations – some attempt to quantify this behavior should be undertaken, either as part of this study or a future study. It would be useful to compare dyad relatedness and probable kinship relationships with an overall index of pairwise association among individuals or with an index of the rate of nursing or other care-type behaviors, in order to determine the importance of alloparental care within the group.

We agree with the reviewer. This will be part of next steps of our study.

595-599: This statement is unclear, in part because organisational complexity hasn't yet been defined. Please describe why alloparental care is more consistent with organizational complexity. Please also clarify the link you are trying to make, between the positive phylogenetic signal of alloparental care in mammals, and sperm whales having a relational complexity.

Here again, following the remark 5 of reviewer 1, we removed from our manuscript all the analysis of the average r value of the Irene's group, including this paragraph (593-599).

663-664: Again, this statement is strong given the sample size, and lack of sample coverage from other social groups in the population. Microsatellite markers indicated low within-group relatedness compared to other studies. Several of the adult females (e.g. Vanessa, Claire) have no relationship to other animals in the group. These findings do not support “strict matrilineality” as stated here.

In the literature, social groups can be considered matrilineal when most females (or all offspring) remain, for life, with their mothers and other close female relatives. Taken together, the observation of the same individuals over a long time period, the mtDNA haplotype homogeneity and the number of first- and second-degree relationships are in agreement with this definition and have highlighted the matrilineality of this social group.

697: Replace “most” with “more” – many marine species exhibit socially transmitted cultural behaviors and socially driven genetic structure.

Done

Figure 2: The figure caption indicates that mitochondrial haplotypes are displayed, but they are not shown in the figure.

Thanks for noticing. This part of the sentence has been removed

Appendix C

Associate Editor Comments to Author:

The reviewers are largely happy with the scientific/analytical changes you have implemented, but there are a number of linguistic/typographic changes recommended that would improve the work. We'd like you to take advantage of this opportunity to tweak the language and punctuation to improve the readability - you might benefit from the use of a language editing service such as those at <https://royalsociety.org/journals/authors/benefits/language-editing/>. We'll look forward to reading your final version soon.

Dear associate editor,

We carefully read your remarks, and those from the reviewers.

We answered all the questions, and made the modifications asked by the reviewers in the text.

We described the modifications made in blue in the following paragraphs, and they are apparent in the file "Sarano et al V2 with correction marks".

This file was then edited by a language editing service (the certificate is joined).

"Sarano et al with English editing" is the final clean version, incorporating all the changes made.

Sincerely yours,

Jean-Luc Jung

Reviewer comments to Author:

Reviewer: 1

Comments to the Author(s)

The manuscript is improved from the first draft; however, there is still quite a bit of room for improvement, although perhaps it is more cosmetic this time. I found in some parts, particularly the Abstract, but also elsewhere, the work was over-sold and the wording should be toned down. Additionally, I think the context of our knowledge of sperm whale social structure is not represented accurately in the Introduction, which provides a false context for this paper. Other than those, I have several more specific suggestions to improve the paper, as described below.

1. Line 48-49: I think it is overselling it to call the sampling technique "innovative" (it is just collecting skin that is seen floating in the water after the animals pass, after all). I suggest changing this to "using underwater observations, individual-specific identification, and genetic analyses based on mitochondrial sequencing and microsatellite profiling."

The sentence has been changed to

"We studied a stable social group of sperm whales off Mauritius, using underwater observations, individual-specific identification, non-invasive sampling and genetic analyses based on mitochondrial sequencing and microsatellite profiling."

2. Lines 53-55: This statement is incorrect. Konrad et al. (2018) did the same thing for multiple groups in their paper (some of which were completely sampled).

"For the first time" has been removed

3. Line 57: I think the word “adoption” here could be misleading. Technically, this would mean that a mother nurses and raises a calf that is not her own (which has been reported in whales). But it is not clear that that happened here, and it would be easy for readers to get confused. I would use clearer wording, perhaps “We highlight a likely case of an unrelated female being integrated into a social unit, presenting a mtDNA...”

The sentence has been modified, as asked

4. Lines 81-83: I think “strongly dependent on kin relationships” is an over-statement. For some of these species it is definitely true, but I think the jury is still out on the importance of relatedness for some of them (e.g., long-finned pilot whales).

“Strongly” has been replaced by “often”

5. Line 91: It would be helpful if the authors define what is meant by a “fission-fusion model” so that all of the readers are on the same page.

A sentence has been added to define fission-fusion models

6. Line 104: This isn’t quite right. These aren’t “genetically distinct” populations. It is true that maternally-directed site-fidelity in baleen whales leads to distinct structure of mitochondrial haplotypes between whales in different feeding grounds. But often whales from different feeding grounds all breed together in the winter. Thus, the feeding groups represent seasonal population structure, but certainly not “genetically distinct populations”.

In the North Pacific, humpback whales present genetic differences between both breeding and feeding grounds (Baker et al, 2013, MEPS, 494, 291-306). In Russian waters, Richard et al. (2018, J Hered, 7, 724-734) highlighted that the two groups of humpback whales feeding in two close sites can be distinguished on the basis on mitochondrial haplotype frequencies, and that, thanks to a comparison with the results of Baker et al. (2013), they were also almost certainly breeding in different locations, in the east of the North Pacific for one group, and in the west for the other group. We would therefore like to keep the sentence as it is. If the sentence must absolutely be changed, “genetically” can simply be removed.

7. Lines 171-172: The statement that “all samples used in these studies were biopsies...” is not true. Many are based on sloughed skin.

See answer to the following remark

8. Lines 173-177: This doesn’t provide an accurate picture of the studies underlying our current knowledge of sperm whale social structure. It is very hand-wavy and inaccurate. This may be true for some populations, but not for others. Painting these studies in this way sets them up as a straw-man, presumably to make this current study look better. This does not seem appropriate.

It is absolutely not our intention to lower in any way the quality and significance of previous studies made by others. We totally agree with the reviewer, that some populations are very well known, in particular thanks to few very long terms and particularly relevant studies (that we cite several times in our manuscript). We just wanted to highlight the fact that our

underwater sampling protocol made on individually-identified sperm whales can strongly help for studies like the one that we presented here.

As this paragraph can be misunderstood, we have removed it.

9. Lines 222-224: I think it is over-selling it. Although perhaps true for this particular geographical region, social structure in sperm whales has been well-studied in other places. This sentence also does not add anything to the paper.

We have changed the order of the words in the sentence, Indian Ocean is now in the first part of the sentence, in order to reinforce the localized area of our study.

10. Lines 292-306: It is a bit weird that some description of the mitochondrial sequencing and microsatellite genotyping are provided here, but then other aspects of these procedures are described separately, under their own subheadings. It would be clearer if the text from here was moved to its appropriate subheading, so that all of the information describing each procedure was kept together. Also on line 292 it states that 774 bp of the control region was amplified, but on line 310 it says 638. Why the difference?

Two paragraphs have been moved.

774 bp is the length of the amplified fragment, 638 bp is the length of the sequence determined on all samples.

11. Line 356: More information should be provided about how this error rate of 2.2% was calculated. Is it per allele, per locus, per genotype? Exactly how many discrepancies were identified? The authors should be very explicit in this explanation. I had this comment last time, and the authors address it in their response, but it should be added to the manuscript.

The calculation has been added in the text (“38 alleles incorrect among 1708 scored”)

12. Lines 512-523: For adult females that were already adult when the study started, and that showed parent-offspring relatedness with each other, how did they authors decide who was the mother and who was the offspring (since all were adult upon the initiation of the study). This should be clearly explained.

This is now explained in the text, this new paragraph has been added.

« Four of the first-degree relationships each involved two adult females. Specifically, (i) Mystère and Irène shared such a relationship, which may reflect either a full-sibling one (which we retained as more probable) or a mother-offspring one. (ii) Germiné was a juvenile in 2013 (her teeth were still emerging), so Issa has to be her mother. (iii) The case was more complex to decipher for Dos-Calleux, Lucy, and Mina—but sperm whale adult females rarely give birth after 40 years of age⁽¹³⁾, which makes it highly unlikely that the mother of Dos-Calleux could also be that of a young sperm whale born in 2018 (e.g., Daren and Ali). The most probable relationship is therefore the one conveyed in Figure 3, where Dos-Calleux is the mother of both Mina and Lucy ».

13. Figure 2: I think the diagram is an intuitive and informative way to visualize the putative relationships. However, my concern is that they could be interpreted as “fact”, whereas there is a lot of variation and uncertainty around relatedness estimates. I think this could be alleviated by just including a sentence in the caption saying something like “This diagram was constructed to be consistent with the analyses conducted; however, there is uncertainty around relatedness estimates and thus around some of these relationships”.

We agree. A sentence has been added in the caption.

14. Line 645-646: The authors say that Caroline “rarely took care of” Alexander, even though she was his mom. This seems to strong. Do the authors mean that she was “rarely associated” with him? Those could be two very different things, and my guess is that the authors cannot say anything about how good of a mother she is. Similarly in line 650: I would suggest removing “taking care of” and replace it with what the actually observed (e.g., “associated with”).

The sentence has been modified

15. I think the map (Figure S1) should be in the paper rather than in the supplementary material.

Done

Reviewer: 2

Comments to the Author(s)

Sarano et al. have done an excellent job responding to comments from both reviewers. I have a few additional minor comments below, but otherwise look forward to seeing this manuscript in press.

Line 81: add scientific names for long and short-finned pilot whales

Done

Line 200: Please add here how many individuals have been photo identified in the population, whether there is an abundance estimate for the local population, and if so what it is.

A sentence has been added about the local population in the preceding paragraph. Around one hundred sperm whales have been photo-identified in the area by authors of the ref 52.

Line 421: should this be “<=” ?

Yes, this has been modified line 426.

Appendix D

Jean-Luc Jung
ISYEB
MNHN - UBO

Brest, January the 7th

Dear Associate editor,

Thank for your comments.

We submit today a new version of our manuscript. We were asked to make two minor changes in the text and to make the microsatellite dataset available.

In this new version, the two modifications asked by the reviewer have been made in the text, following the wording proposed by the reviewer.

We have also deposited the microsatellite dataset in dryad. The accession link is indicated in the publication, under the paragraph “data accessibility” doi:10.5061/dryad.bcc2fqzbk (temporary link <https://datadryad.org/stash/share/s9ppia0XII8a7FW9CSsfhFZSJKYaMzfzYN3eTPRvIFg>.

)

We thank once again the reviewer for his work on our publication, and we hope that this new version of our manuscript will be suitable for publication in RSOS.

Sincerely yours,